# Efficient Alternating Minimization with Applications to Weighted Low Rank Approximation

**Zhao Song**
Simons Institute for the Theory of Computing, UC Berkeley
magic.linuxkde@gmail.com

**Mingquan Ye**
University of Illinois Chicago
mye9@uic.edu

**Junze Yin**
Rice University
jy158@rice.edu

**Lichen Zhang**
MIT CSAIL
lichenz@csail.mit.edu

## Abstract

Weighted low rank approximation is a fundamental problem in numerical linear algebra, and it has many applications in machine learning. Given a matrix $M \in \mathbb{R}^{n \times n}$, a non-negative weight matrix $W \in \mathbb{R}_{\geq 0}^{n \times n}$, a parameter $k$, the goal is to output two matrices $X, Y \in \mathbb{R}^{n \times k}$ such that $\|W \circ (M - XY^\top)\|_F$ is minimized, where $\circ$ denotes the Hadamard product. It naturally generalizes the well-studied low rank matrix completion problem. Such a problem is known to be NP-hard and even hard to approximate assuming the Exponential Time Hypothesis (Gillis & Glineur, 2011; Razenshteyn et al., 2016). Meanwhile, alternating minimization is a good heuristic solution for weighted low rank approximation. In particular, Li et al. (2016) shows that, under mild assumptions, alternating minimization does provide provable guarantees. In this work, we develop an efficient and robust framework for alternating minimization that allows the alternating updates to be computed approximately. For weighted low rank approximation, this improves the runtime of Li et al. (2016) from $\|W\|_0 k^2$ to $\|W\|_0 k$ where $\|W\|_0$ denotes the number of nonzero entries of the weight matrix. At the heart of our framework is a high-accuracy multiple response regression solver together with a robust analysis of alternating minimization.

## 1 Introduction

Given a matrix $M \in \mathbb{R}^{n \times n}$, the low rank approximation problem with rank $k$ asks us to find a pair of matrices $\widetilde{X}, \widetilde{Y} \in \mathbb{R}^{n \times k}$ such that $\|M - \widetilde{X}\widetilde{Y}^\top\|_F$ is minimized over all rank $k$ matrices $X$ and $Y$, where $\|\cdot\|_F$ is the Frobenius norm of a matrix. Finding a low rank approximation efficiently is a core algorithmic problem that is well studied in machine learning, numerical linear algebra, and theoretical computer science. The exact solution follows directly from singular value decomposition (SVD): let $M = U\Sigma V^\top$ and set $\widetilde{X} = U_k \sqrt{\Sigma_k}, \widetilde{Y} = V_k \sqrt{\Sigma_k}$, i.e., picking the space spanned by the top-$k$ singular values and corresponding singular vectors. Faster algorithms utilizing linear sketches can run in input sparsity time (Clarkson & Woodruff, 2013). In addition to the standard model and Frobenius norm, low rank approximation has also been investigated in distributed setting (Boutsidis et al., 2016), for entrywise $\ell_1$ norm (Song et al., 2017) and for tensors (Song et al., 2019c).

In practice, it is often the case that some entries of $M$ are more important than others and some entries can be completely ignored, so it's natural to look for a *weighted* low rank approximation. More specifically, given a target matrix $M \in \mathbb{R}^{n \times n}$ and a non-negative weight matrix $W \in \mathbb{R}_{\geq 0}^{n \times n}$, the goal is to find $\widetilde{X}, \widetilde{Y} \in \mathbb{R}^{n \times k}$ with $\|W \circ (M - \widetilde{X}\widetilde{Y}^\top)\|_F$ minimized, where $\circ$ is the Hadamard product of two matrices. The formulation of weighted low rank approximation covers many interesting matrix problems, for example, the classic low rank approximation can be recovered by setting $W = \mathbf{1}_n \mathbf{1}_n^\top$ and the matrix completion problem (Jain et al., 2013) is by observing a subset of entries of $M$, equivalent to picking $W$ as a Boolean matrix. In addition to its theoretical importance, weighted low rank approximation also has a significant practical impact in many fields, such as natural language

processing (Pennington et al., 2014; Arora et al., 2016; Hsu et al., 2022), collaborative filtering (Srebro & Jaakkola, 2003; Koren et al., 2009; Lee et al., 2013; Chen et al., 2015), ecology (Robin et al., 2019; Kidzinski et al., 2022), chromatin conformation reconstruction Tuzhilina et al. (2022) and statistics (Wentzell et al., 1997; Markovsky et al., 2006).

Algorithmic study for weighted low rank approximation dates back to Young (1941). On the computational hardness front, Gillis & Glineur (2011) has shown that the general weighted low rank approximation is NP-hard even if the ground truth matrix is rank 1. The hardness is further enhanced by Razenshteyn et al. (2016) by showing that assuming the Random Exponential Time Hypothesis, the problem is hard to approximate beyond a constant factor. Despite its hardness, many heuristic approaches have been proposed and witnessed many successes. For example, Shpak (1990) implements gradient-based algorithms, while Lu et al. (1997); Lu & Antoniou (2003) use the alternating minimization framework. Srebro & Jaakkola (2003) develops algorithm based on expectation-maximization (EM). Unfortunately, all these approaches are without provable guarantees. Razenshteyn et al. (2016) is the first to provide algorithms with theoretical guarantees. They propose algorithms with parameterized complexity on different parameters of $W$, such as the number of distinct columns or low rank. In general, these algorithms are not polynomial which is also indicated by their lower bound results. Ban et al. (2019) subsequently studies the weighted low rank approximation problem with regularization, and they manage to obtain an improved running time depending on the statistical dimension of the input, rather than the rank. When one relaxes to a bi-criteria solution with additive error guarantees, Bhaskara et al. (2021) provides a greedy algorithm. Whenever all entries of the weight matrix are nonzero, Dai shows that it is possible to convert the additive error to multiplicative (Dai, 2023).

How to bypass the barrier of Razenshteyn et al. (2016) while still getting provable guarantees? Li et al. (2016) draws inspirations from matrix completion literature and views the problem as a low rank matrix recovery problem: suppose the matrix $M \in \mathbb{R}^{n \times n}$ is a noisy, full-rank observation that can be decomposed into $M = M^* + N$ where $M^*$ is the rank-$k$ ground truth and $N$ is the rank-$(n-k)$ noise matrix. They then analyze the performance of alternating minimization when 1). the ground truth is incoherent, 2). weight matrix has a spectral gap to all-1's matrix, and 3). weight matrix is non-degenerate. Under these assumptions, they show that the alternating minimization algorithm provably finds a pair of matrices $\widetilde{X}, \widetilde{Y} \in \mathbb{R}^{n \times k}$ such that $\|M - \widetilde{X}\widetilde{Y}^\top\| \leq O(k) \cdot \|W \circ N\| + \epsilon$, where $\| \cdot \|$ is the spectral norm of a matrix. This provides a solid theoretical ground on why alternating minimization works for weighted low rank approximation.

While the Li et al. (2016) analysis provides a polynomial time algorithm for weighted low rank approximation under certain assumptions, the algorithm itself is still far from efficient. In particular, the alternating minimization framework requires one to solve $O(n)$ different linear regressions exactly per iteration. The overall runtime of their algorithm is $O((\|W\|_0 \cdot k^2 + nk^3) \log(1/\epsilon))$ where $\|W\|_0$ denote the number of nonzero entries in $W$, making it inefficient for practical deployment. Moreover, their analysis is non-robust, meaning that it cannot account for any error at each step. This is in drastic contrast with practice, where floating point errors and inexact solvers are used everywhere. In fact, there are good reasons for them to mandate exact regression solvers, as their algorithm only requires $\log(1/\epsilon)$ iterations to converge and any fast but approximate regression solver might break the nice convergence behavior of the algorithm. Hence, we ask the following question:

*Is it possible to obtain a faster and more robust alternating minimization-based algorithm with a similar convergence rate?*

In this paper, we provide a positive answer to this question. Specifically, we show that the alternating updates can be computed in *nearly linear time* each iteration and *polynomially large errors* can be tolerated. Both of these results rely on a fast, randomized and high-accuracy regression solver that uses sketching to compute a preconditioner. We summarize our main result in the following theorem:

**Theorem 1.1** (Informal version of Theorem 4.6). *There is an algorithm (see Algorithm 1) that runs in $\widetilde{O}((\|W\|_0 \cdot k + nk^3) \log(1/\epsilon))$ time and outputs a rank-$k$ matrix $\widetilde{M}$ such that*

$$\|\widetilde{M} - M^*\| \leq O(k\tau) \cdot \|W \circ N\| + \epsilon$$

*where $\tau$ is the condition number of $M^*$ and $\widetilde{O}(\cdot)$ suppresses polylogarithmic factors in $n$ and $k$.*

---

**Algorithm 1** Main Algorithm. The CLIP procedure zeros out rows whose $\ell_2$ norm are large, and the QR procedure computes the QR decomposition of the matrix and outputs the orthonormal factor $Q$.

---

1: **procedure** FASTERWEIGHTLOWRANK($M \in \mathbb{R}^{n \times n}$, $W \in \mathbb{R}^{n \times n}$, $\epsilon$,$k$)                    ▷ Theorem 1.1
2:     $T \leftarrow O(\log(1/\epsilon))$
3:     $\delta_{\text{sk}} \leftarrow 1/\text{poly}(n, T)$
4:     $\epsilon_{\text{sk}} \leftarrow 1/\text{poly}(n, \tau)$                    ▷ $\tau$ is an estimate of the condition number of $M^*$.
5:     $Y_0 \leftarrow \text{RANDOMINIT}(n, k)$ ▷ Initialize $Y_0$ to random Rademacher variables, scaled by $\frac{1}{\sqrt{n}}$.
6:     **for** $t = 1$ to $T$ **do**
7:         $\vec{X}_t \leftarrow \text{FASTMULTIPLEREGRESSION}(M, Y_{t-1}, W, \epsilon_{\text{sk}}, \delta_{\text{sk}})$     ▷ Solve $O(n)$ regressions
    using sparsity of $W$ and Algorithm 2.
8:         $\widehat{X}_t \leftarrow \text{CLIP}(\vec{X}_t)$                              ▷ Clip rows with large $\ell_2$ norms.
9:         $X_t \leftarrow \text{QR}(\widehat{X}_t)$
10:        $\vec{Y}_t \leftarrow \text{FASTMULTIPLEREGRESSION}(M^\top, X_t, W^\top, \epsilon_{\text{sk}}, \delta_{\text{sk}})$
11:        $\widehat{Y}_t \leftarrow \text{CLIP}(\vec{Y}_t)$
12:        $Y_t \leftarrow \text{QR}(\widehat{Y}_t)$
13:    **end for**
14:    **return** $\widetilde{M} \leftarrow \widehat{X}_T Y_{T-1}^\top$
15: **end procedure**

---

**Remark 1.2.** The general structure of our main algorithm (Algorithm 1) is based on the traditional alternating minimization method described in Li et al. (2016). We replace the exact update with an approximate update (lines 7 and 10) based on Algorithm 2, which makes the overall algorithm both faster and more robust. The remainder of the paper is dedicated to presenting a theoretical guarantee for its efficiency and robustness.

**Roadmap.**    In Section 2, we introduce several basic notations and definitions which we will use throughout this paper. In Section 3, we give a brief overview of our techniques. In Section 4, we present our main result. In Section 5, we give a conclusion for this paper.

## 2    PRELIMINARY

In Section 2.1, we introduce the basic notation used in this paper. In Section 2.2, we present the background of the sketching technique, including the SRHT matrix and oblivious subspace embedding. In Section 2.3, we present the mathematical background and assumptions related to the weighted low rank approximation problem.

### 2.1    NOTATION

Let $n, m$ be arbitrary positive integers. We define a set $[n]$ as $\{1, 2, \cdots, n\}$. We use $\mathbb{R}, \mathbb{R}^m, \mathbb{R}^m_{\geq 0}$, and $\mathbb{R}^{n \times m}$ to denote the sets containing all the real numbers, $m$-dimensional vectors with real entries, $m$-dimensional vectors with non-negative real entries, and $n \times m$ matrices with real entries.

Let $x \in \mathbb{R}^m_{\geq 0}$ and $w \in \mathbb{R}^m_{\geq 0}$. Let $i \in [m]$. Let $x_i \in \mathbb{R}$ represent the $i$-th entry of $x$. We use $\sqrt{x} \in \mathbb{R}^m$ to represent a vector satisfying $(\sqrt{x})_i = \sqrt{x_i}$. We define $\|x\|_w := (\sum_{i=1}^n w_i x_i^2)^{1/2}$.

Let $A, W$ be two arbitrary matrices in $\mathbb{R}^{n \times m}$. Let $i \in [n]$ and $j \in [m]$. We use $A_{i,:} \in \mathbb{R}^m$ to represent a column vector that is equal to the $i$-th row of $A$ and $A_{:,j} \in \mathbb{R}^n$ represent a column vector that is equal to the $j$-th column of $A$. $A_{i,j} \in \mathbb{R}$ represents a entry of $A$, located at the $i$-th row and $j$-th column. $\text{diag}(x) \in \mathbb{R}^{n \times n}$ represents the matrix satisfying $\text{diag}(x)_{i,j} = x_i$ if $i = j$ and $\text{diag}(x)_{i,j} = 0$ if $i \neq j$. $\text{nnz}(A)$ represents the number of nonzero entries of $A$.

Suppose that $n \geq m$. We denote the spectral norm of $A$ as $\|A\| = \sup_{x \in \mathbb{R}^m} \|Ax\|_2 / \|x\|_2$, denote the Frobenius norm of $A$ as $\|A\|_F$, which is equal to $(\sum_{i=1}^n \sum_{j=1}^m A_{i,j}^2)^{1/2}$, and denote $\|A\|_{\infty,1}$ as $\max\{\max_{i \in [n]} \|A_{i,:}\|_1, \max_{j \in [m]} \|A_{:,j}\|_1\}$.

Further, let $U\Sigma V^\top$ be the singular value decomposition (SVD) of $A$. Then, we have $U \in \mathbb{R}^{n \times m}$ and $\Sigma, V \in \mathbb{R}^{m \times m}$, where $U, V$ have orthonormal columns and $\Sigma$ is a non-negative diagonal matrix. The Moore-Penrose pseudoinverse of a matrix $A$ is $A^\dagger = V\Sigma^{-1}U^\top$. If $\Sigma$ is a sorted diagonal matrix and $\sigma_1, \cdots, \sigma_m$ represent the diagonal entries of $\Sigma$, then we use $\sigma_i$ to represent the $i$-th singular value of $A$, namely $\sigma_i(A)$. We define $\sigma_{\min}(A) := \min_x \|Ax\|_2/\|x\|_2$ and $\sigma_{\max}(A) := \max_x \|Ax\|_2/\|x\|_2$.

Now, we suppose that $m = n$, namely $A, W \in \mathbb{R}^{n \times n}$. We define $\|A\|_W := \sqrt{\sum_{i=1}^n \sum_{j=1}^n W_{i,j}A_{i,j}^2}$. $W \circ A$ is a matrix whose entries are defined as $(W \circ A)_{i,j} := W_{i,j}A_{i,j}$. We define $D_{W_i} := \mathrm{diag}(W_{:,i})$. If $A$ is invertible, then the true inverse of $A$ is denoted as $A^{-1}$ and $\|A\| = \sigma_{\min}(A^{-1})$. If $A$ is symmetric, then we define as $U\Lambda U^\top$ the eigenvalue decomposition of $A$, where $\Lambda$ is a diagonal matrix. Let $\lambda_1, \cdots, \lambda_n$ represent the entries on diagonal of $\Lambda \in \mathbb{R}^{n \times n}$. $\lambda_i$ is called the $i$-th eigenvalue, namely $\lambda_i(B)$. Furthermore, the eigenvalue and the singular value satisfy $\sigma_i^2(A) = \lambda_i(A^\top A)$. Given two $n \times n$ real symmetric matrices $A$ and $B$, we use $A \preceq B$ to denote the matrix $B - A$ is positive semidefinite, i.e., for any $x \in \mathbb{R}^n$, $x^\top(B - A)x \geq 0$.

## 2.2 SKETCHING

An important algorithmic subroutine is the Subsampled Randomized Hadamard Transform SRHT:

**Definition 2.1** (SRHT (Lu et al., 2013))**.** The SRHT matrix of size $m \times n$ is the following matrix: $S = \frac{1}{\sqrt{m}}PHD$, where $D \in \mathbb{R}^{n \times n}$ is a diagonal matrix with diagonal being Rademacher random variables, $H \in \mathbb{R}^{n \times n}$ is the Hadamard matrix and $P \in \mathbb{R}^{m \times n}$ is a row sampling matrix that samples $m$ rows with replacement.

The key property we would like to leverage from SRHT is the *subspace embedding property*:

**Definition 2.2** (Oblivious subspace embedding (Sarlos, 2006))**.** Let $n, d$ be positive integers and $\epsilon, \delta \in (0, 1)$ be parameters, we say a distribution $\Pi$ over $m \times n$ real matrices satisfy $(\epsilon, \delta, n, d)$-oblivious subspace embedding (OSE) if for any fixed matrix $A \in \mathbb{R}^{n \times d}$ and $S \sim \Pi$, with probability at least $1 - \delta$, we have for any $x \in \mathbb{R}^d$,

$$(1 - \epsilon)\|Ax\|_2 \leq \|SAx\|_2 \leq (1 + \epsilon)\|Ax\|_2.$$

Via standard matrix concentration inequalities such as matrix Chernoff bound (see e.g. Rudelson (1999); Ahlswede & Winter (2002)), one can show SRHT with $m = O(\epsilon^{-2}d\log^2(n/\delta))$ satisfying $(\epsilon, \delta, n, d)$-OSE. Moreover, since $H$ is a Hadamard matrix, applying $S$ to an $n$-dimensional vector can be done in $O(n \log n)$ using FFT. Thus, computing $SA$ takes $O(nd \log n)$ time.

## 2.3 BACKGROUND ON WEIGHTED LOW RANK APPROXIMATION

The weighted low rank approximation can be treated as a generalization of the noisy matrix completion problem, where the goal is to recover a target matrix $M \in \mathbb{R}^{n \times n}$ from a few observations (sublinear in $n^2$) where the weight is chosen as a Boolean matrix $P_\Omega \in \mathbb{R}^{n \times n}$. It is hence natural to impose and generalize assumptions from matrix completion if we would like to obtain any provable guarantees. Following Li et al. (2016), we make three assumptions and we will justify them one by one.

**Assumption 2.3.** Given a noisy, possibly higher-rank observation $M \in \mathbb{R}^{n \times n}$ such that $M = M^* + N$, where $M^*$ is the rank-$k$ ground truth we want to recover and $N$ is the noise matrix. We assume:

1. $M^*$ is $\mu$-incoherent: Let $M^* = U\Sigma V^\top$ be its SVD, we assume

$$\max\{\|U_{i,:}\|_2^2, \|V_{i,:}\|_2^2\}_{i=1}^n \leq \frac{\mu k}{n}.$$

We use $\tau$ to denote the condition number of $M^*$: $\tau = \sigma_{\max}(M^*)/\sigma_{\min}(M^*)$.

2. Weight $W$ has a $\gamma$-spectral gap to all-1's matrix:

$$\|W - \mathbf{1}_n\mathbf{1}_n^\top\| \leq \gamma n.$$

3. Weight $W$ is $(\alpha, \beta)$-bounded: Let $M^* = U\Sigma V^\top$ be its SVD, we assume for any $i \in [n]$ and $0 < \alpha \leq 1 \leq \beta$,

$$\alpha I \preceq U^\top D_{W_i} U \preceq \beta I,$$
$$\alpha I \preceq V^\top D_{W_i} V \preceq \beta I.$$

Assumption 1 states that the largest row norms of the left and right singular factors should not be too far away from the average. Such matrix incoherence assumption has been very standard in the context of matrix completion (Candès & Recht, 2012) as it effectively eliminates the degenerate case where the ground truth $M^*$ has very weak signals. Consider the extreme case where $M^* = e_1 e_1^\top$, in such a scenario, if the weight $W$ is rather uniform over all entries and $N$ is a dense noise matrix with its first entry has a small magnitude compared to other entries, then recovering $M^*$ will be next to impossible. The incoherence assumption makes sure that the row and column space of $M^*$ are spread over coordinates. Incoherence is also commonly observed in practice (Mohri & Talwalkar, 2011).

Assumption 2 is a natural generalization of the *random sampling assumption* for matrix completion (Jain et al., 2013; Hardt, 2014). In particular, if $W$ is a Boolean matrix where each row has $\Omega(\log n)$ entries chosen uniformly at random, then $\gamma = O(\frac{1}{\sqrt{\log n}})$. Generalize to a non-negative weight setting, it also bounds the largest possible magnitude of any entry in $W$ to avoid degeneracy.

Assumption 3 is also best understood when $W$ is a Boolean matrix, so that $D_{W_i}$ selects subset of rows of $U$ and $V$, and the condition essentially reduces to Assumption A2 of Bhojanapalli & Jain (2014). It is a strengthening and weighted generalization of the strong incoherence property as it directly implies the assumption in Candès & Tao (2010), which is necessary for matrix completion.

Having justified the assumptions we impose on the ground truth and the weight, we are in the position to state the weighted low rank approximation problem.

**Problem 2.4.** *Let $M \in \mathbb{R}^{n \times n}$ be a noisy, higher-rank matrix with $M = M^* + N$ where $M^*$ is the rank-$k$ ground truth and $N$ is a higher-rank noise matrix. Let $W \in \mathbb{R}_{\geq 0}^{n \times n}$ be a non-negative weight matrix. Suppose both $M^*$ and $W$ satisfy Assumption 2.3. The goal is to find a rank-$k$ matrix $\widetilde{M} \in \mathbb{R}^{n \times n}$ such that*

$$\|\widetilde{M} - M^*\| \leq \delta \cdot \|W \circ N\| + \epsilon$$

*by observing the matrix $W \circ M$.*

When $W$ is a Boolean matrix, Problem 2.4 reduces to the noisy matrix completion problem where one needs to recover the rank-$k$ ground truth by observing a few entries of a higher-rank noisy matrix.

## 3 TECHNIQUE OVERVIEW

In this section, we provide a preliminary overview of the techniques we use in this paper. Before diving into our algorithm and analysis, let us first review the algorithm of Li et al. (2016). At each iteration, the algorithm alternates by solving two weighted multiple response regressions: starting with an initial matrix $Y$, it tries to find a matrix $X \in \mathbb{R}^{n \times k}$ that minimizes $\|W \circ (M - XY^\top)\|_F^2$, then they zero out the rows of $X$ with large $\ell_2$ norms and use the QR factor of $X$ to proceed. Then, they alternate and solve $\min_{Y \in \mathbb{R}^{n \times k}} \|W \circ (M - XY^\top)\|_F^2$ given the new $X$. After properly zeroing out large rows and QR, the algorithm proceeds to the next iteration. The main runtime bottleneck is to solve the weighted multiple response regression per iteration.

Following the trend of low rank approximation (Clarkson & Woodruff, 2013) and fixed parameter tractable algorithm for weighted low rank approximation (Razenshteyn et al., 2016), it is natural to consider using sketching to speed up the multiple response regression solves. Let us consider

$$\min_{Y \in \mathbb{R}^{n \times k}} \|W \circ (M - XY^\top)\|_F^2. \tag{1}$$

Let $D_{\sqrt{W_i}}$ denote the $n \times n$ diagonal matrix that puts $\sqrt{W_i}$ on the diagonal, where $W_i$ is the $i$-th column of $W$. It is not hard to verify that (1) can be cast into $n$ linear regressions (see details in Claim C.1), each of which is in the form of

$$\min_{y \in \mathbb{R}^k} \|D_{\sqrt{W_i}} M_{:,i} - D_{\sqrt{W_i}} X y\|_2^2.$$

To solve these regressions faster, one can pick a random sketching matrix $S \in \mathbb{R}^{s \times n}$ where $s = O(\epsilon_0^{-2} k)$ and instead solve

$$\min_{y \in \mathbb{R}^k} \|SD_{\sqrt{W_i}} M_{:,i} - SD_{\sqrt{W_i}} Xy\|_2^2.$$

By picking a sparse sketching matrix $S$ (Nelson & Nguyên, 2013), the above regression can be solved in $\widetilde{O}(\epsilon_0^{-1} \operatorname{nnz}(X) + \epsilon_0^{-2} k^3)$ time with high probability, and the output solution $y$ has cost at most $(1 + \epsilon_0) \cdot \text{OPT}$ where OPT is the optimal regression cost. Aggregate over $n$ regressions, this gives an $\widetilde{O}(\epsilon_0^{-1} n \cdot \operatorname{nnz}(X) + \epsilon_0^{-2} n k^3)$ time per iteration (see Lemma C.6).

This approach, however, has several drawbacks that make it infeasible for our application. The first is the error guarantee of such approximates regression solves. Essentially, we compute a matrix $\widetilde{Y} \in \mathbb{R}^{n \times k}$ such that

$$\|W \circ (M - X\widetilde{Y}^\top)\|_F^2 \le (1 + \epsilon_0) \cdot \min_{Y \in \mathbb{R}^{n \times k}} \|W \circ (M - XY^\top)\|_F^2,$$

in other words, the approximate solution $\widetilde{Y}$ provides a relative *forward error*. Unfortunately, the forward error is much less helpful when we want to analyze how close $\widetilde{Y}$ is to the optimal solution $Y$, i.e., the *backward error*. It is possible to convert forward error to backward error at the expense of dependence on other terms such as the cost of the regression and the spectral norm of $X^\dagger$, the pseudo-inverse of $X$. To cancel out the effect of these extra terms, we will have to set the error parameter $\epsilon_0$ to be very small, thus, a polynomial dependence on $\epsilon_0^{-1}$ in the running time is unacceptable.

This motivates us to design a fast and high precision regression solver whose $\epsilon_0$ dependence is $\log(1/\epsilon_0)$ (see Lemma C.10). Given an algorithm that produces an $(1 + \epsilon_0)$ relative forward error of regression in $\log(1/\epsilon_0)$ iterations, we can set $\epsilon_0$ to inverse proportionally to $\text{OPT} \cdot \|(W \circ X)^\dagger\|$. As the spectral norm of $(W \circ X)^\dagger$ is polynomially bounded, this incurs an extra $\log n$ term in the runtime. It remains to devise a regression solver with such runtime behavior. Our approach is to use the sketch as a preconditioner: we pick a dense sketching matrix $S \in \mathbb{R}^{s \times n}$ with $s = \widetilde{O}(k)$ rows such that for any $k$-dimensional vector $x$, $\|Sx\|_2 = (1 \pm O(1)) \cdot \|x\|_2$. We then apply $S$ to $D_{\sqrt{W_i}} X$ to form a short and fat matrix and compute the QR decomposition of this matrix. It turns out that the right QR factor of $SD_{\sqrt{W_i}} X$ is a good preconditioner to $D_{\sqrt{W_i}} X$. We then use $S$ to find a constant approximation to the regression problem and utilize it as a starting point. The algorithm then iteratively performs gradient descent to optimize towards an $\epsilon_0$-approximate solution. Overall, such an algorithm takes $\log(1/\epsilon_0)$ iterations to converge, and each iteration can be implemented in $\widetilde{O}(nk)$ time. Plus the extra $\widetilde{O}(nk + k^3)$ time to compute the initial solution, this yields an algorithm that runs in $\widetilde{O}((nk + k^3) \log(1/\epsilon_0))$ time to compute an $\epsilon_0$ forward error solution. Note here we sacrifice the input sparsity time in exchange of a sketching matrix that works with high probability. This also accounts for the fact that both $X$ and $Y$ are quantities changed across iterations and the sparsity cannot be controlled.

The runtime can be further improved by leveraging the sparsity of the weight matrix $W$. Again, consider the regression $\min_{y \in \mathbb{R}^k} \|D_{\sqrt{W_i}} M_{:,i} - D_{\sqrt{W_i}} Xy\|_2^2$, if $W_i$ only has a few nonzero entries, then the diagonal matrix $D_{\sqrt{W_i}}$ will effectively zero out most rows of $X$ and entries of $M_{:,i}$. This means that we are solving a regression of size $O(\|W_i\|_0 k)$ instead of $O(nk)$. As we iterate through all $n$ regressions, the total instance size is then $O(\sum_{i=1}^n \|W_i\|_0 k) = O(\|W\|_0 k)$, and we can effectively solve these regressions in an overall $\widetilde{O}((\|W\|_0 k + nk^3) \log(1/\epsilon_0))$ time. We note that in matrix completion, $\|W\|_0$ is oftentimes $\widetilde{O}(n \operatorname{poly}(k))$, making it much smaller than $O(n^2)$ and an algorithm that exploits its sparsity is therefore much more valuable.

We want to remark that our high precision and dense regression solver not only works for weighted low rank approximation, but for any alternating minimization frameworks that require one to solve $O(1)$ multiple response regressions per iteration. Due to the good error dependence, the overall $\log(1/\epsilon)$ convergence is well-preserved, even though each iteration is only solved approximately. We believe this high precision solver will also find its use in problems like (low rank) matrix sensing and tasks in which backward error for multiple response regression is required.

In addition to our high-accuracy, high probability solver, we also devise a robust analytical framework for alternating minimization, which is the core to enable us with fast approximate solvers. In particular, we show that if we only output a matrix $\widetilde{Y}$ that is close to the exact regression solution $Y$ in the

spectral norm, then the alternating minimization still converges to the fixed point $\|W \circ N\|$ with good speed. Our analysis uses a different strategy from Li et al. (2016) where they heavily rely on the closed-form of the regression solution. In contrast, we show that by a clever decomposition of errors, one can accumulate the error caused by approximate solves to the additive $\epsilon$ term and thus polylogarithmically more rounds of the iterative solve suffices to give us good guarantees. When adapting our analysis to noiseless matrix completion (Gu et al., 2024b), we recover their result in both runtime and sample complexity, while offering a much simpler proof.

## 4 OUR RESULTS

In Section 4.1, we analyze the weighted multiple response regression. In Section 4.2, we show that the alternating minimization framework is robust, namely this alternating minimization framework can tolerate the error induced by the approximate solver and error conversion. In Section 4.3, we present the formal version of our main result. Finally, in Section 4.4, we compare our results and contribution with those of prior works.

### 4.1 WEIGHTED MULTIPLE RESPONSE REGRESSION

One of our cornerstone results is a novel adaptation of a high-accuracy regression solver based on sketching. Its root can be perhaps traced back to Rokhlin & Tygert (2008), and our two new insights are: 1). This type of high-accuracy regression solvers can also be generalized to *weighted case*, where the design matrix and target vector are scaled by some non-negative weights. 2). We can convert the error on the *cost* of the regression to the error on the *solution*. This step is crucial, as to bridge the gap between our fast, approximate solves and the exact solutions used in Li et al. (2016), it is essentially to quantify the difference between solutions.

**Lemma 4.1.** *Let $A \in \mathbb{R}^{n \times d}$, $b \in \mathbb{R}^n$ and $w \in \mathbb{R}_{\geq 0}^d$. Let $\epsilon \in (0, 0.1)$ be an accuracy parameter and $\delta \in (0, 1)$ be the failure probability. Suppose $\mathcal{T}(n, d, \epsilon, \delta)$ is the runtime of a black-box regression solver that produces a vector $x' \in \mathbb{R}^d$ such that*

$$\|Ax' - b\|_2 \leq (1 + \epsilon) \min_{x \in \mathbb{R}^d} \|Ax - b\|_2$$

*with probability at least $1 - \delta$. Then, there exists an algorithm that runs in time*

$$O(\mathrm{nnz}(A)) + \mathcal{T}(n, d, \epsilon, \delta)$$

*and outputs a vector $x' \in \mathbb{R}^d$ such that with probability at least $1 - \delta$,*

$$\|Ax' - b\|_w \leq (1 + \epsilon) \min_{x \in \mathbb{R}^d} \|Ax - b\|_w.$$

The proof relies on a simple observation: the weights could be applied by scaling rows of $A$ and entries of $b$, which in turn could be implemented in nearly linear time. This simple reduction allows us to deploy a fast off-the-shelf regression solver for weighted regression. To facilitate the analysis, we also require a conversion from the regression cost to how close our approximate solution is to the optimal solution.

**Lemma 4.2.** *Let $A \in \mathbb{R}^{n \times d}$ with $n \geq d$ and full rank, $b \in \mathbb{R}^n$ and let $x_{\mathrm{OPT}}$ be the exact solution to the regression problem $\min_{x \in \mathbb{R}^d} \|Ax - b\|_2$. Suppose there exists a vector $x' \in \mathbb{R}^d$ with*

$$\|Ax' - b\|_2 \leq (1 + \epsilon)\|Ax_{\mathrm{OPT}} - b\|_2,$$

*then we have*

$$\|x' - x_{\mathrm{OPT}}\|_2 \leq O(\sqrt{\epsilon}) \cdot \frac{1}{\sigma_{\min}(A)} \cdot \|Ax_{\mathrm{OPT}} - b\|_2.$$

The conversion from forward to backward error is standard (Price et al., 2017; Gu et al., 2024b), and it means that we will have to set $\epsilon$ to be polynomially small in $\sigma_{\min}(A)$ and the cost of the optimal solution. We combat this issue by employing a high-accuracy regression solver.

The rough idea behind Algorithm 2 is to compute a quick preconditioner using sketching. Let $S \in \mathbb{R}^{m \times n}$ be an SRHT matrix with $m = O(\epsilon_1^{-2} d \log^2(n/\delta))$ rows, it is an $(0.01, \delta, n, d)$-OSE,

---

**Algorithm 2** High precision solver.

---

1: **procedure** HIGHPRECISIONREG($A \in \mathbb{R}^{n \times d}, b \in \mathbb{R}^n, \epsilon \in (0,1), \delta \in (0,1)$)    ▷ Lemma C.10
2:     $\epsilon_1 \leftarrow 0.01$
3:     $m \leftarrow O(\epsilon_1^{-2} \cdot d \log^2(n/\delta))$
4:     Let $S \in \mathbb{R}^{m \times n}$ be an SRHT matrix
5:     Compute QR decomposition of $SA = QR^{-1}$
6:     $x_0 \leftarrow \arg\min_{x \in \mathbb{R}^d} \|SARx - Sb\|_2$
7:     $T \leftarrow C \cdot \log(1/\epsilon)$ for sufficiently large constant $C$
8:     **for** $t = 0 \to T$ **do**
9:         $x_{t+1} \leftarrow x_t + R^\top A^\top (b - ARx_t)$
10:    **end for**
11:    **return** $Rx_T$
12: **end procedure**

---

therefore with high probability, the singular values of $SA$ are close to $A$. The QR decomposition of $SA$ provides an orthonormal basis $Q$ and a non-singular upper triangular matrix $R^{-1}$ which serves as a good preconditioner for $A$. We can then proceed with preconditioned gradient descent using $R$. This procedure is particularly fast because the most time-consuming step is to compute the QR decomposition, but it is performed on an $m \times d$ matrix. Further, $SA$ can be carried out in nearly linear time, and all subsequent steps in gradient descent can be performed in a manner that takes nearly linear time. The property of SRHT also ensures our initial point $x_0$ is a constant approximation of the optimal point, therefore the algorithm converges in $O(\log(1/\epsilon))$ iterations, as desired. For more details, we refer readers to Appendix C.

**Lemma 4.3.** *Given a matrix $A \in \mathbb{R}^{n \times d}$ and a vector $b \in \mathbb{R}^n$, let $\epsilon \in (0, 0.1)$ and $\delta \in (0, 0.1)$, there exists an algorithm that takes time*

$$O((nd \log n + d^3 \log^2(n/\delta)) \log(1/\epsilon))$$

*and outputs $x' \in \mathbb{R}^d$ such that*

$$\|Ax' - b\|_2 \leq (1 + \epsilon) \min_{x \in \mathbb{R}^d} \|Ax - b\|_2$$

*holds with probability $1 - \delta$.*

### 4.2 ROBUSTNESS ANALYSIS FOR APPROXIMATE UPDATE

Now that we have the regression solvers that can compute an approximate update in nearly linear time, we need to show that the alternating minimization framework is robust enough to tolerate the large error induced by the approximate solver. We introduce a generalized incoherence notion.

**Definition 4.4.** Let $A \in \mathbb{R}^{n \times k}$, we define the generalized incoherence of $A$ as $\rho(A) = \frac{n}{k} \cdot \max_{i \in [n]}\{\|A_{i,:}\|_2^2\}$.

As our analysis crucially exploits the interplay between exact and approximate updates, we summarize the notations in the following table to simplify the discussion.

Table 1: Summarization of notations regarding exact and approximate regression solves. By "clipped", we mean zeroing out rows with large $\ell_2$ norms.

| Notation | Meaning |
|---|---|
| $\widetilde{X}$ | Matrix for exact regression solve |
| $\overline{X}$ | Clipped matrix of $\widetilde{X}$ |
| $X$ | QR factor of $\overline{X} = XR$ |
| $\vec{X}$ | Matrix for approximate regression solve |
| $\widehat{X}$ | Clipped matrix of $\vec{X}$ |

**Lemma 4.5.** *Let $Y \in \mathbb{R}^{n \times k}$ be a matrix with orthonormal columns and $\xi$ and $\epsilon_{\mathrm{sk}}$ be parameters and $\Delta_u$ be a parameter depends on $\xi, \epsilon_{\mathrm{sk}}$. Let $\widetilde{X}, \overline{X}, X, \vec{X}$ and $\widehat{X}$ be defined as in Table 1 with the clipping threshold being $4\xi$. Moreover, we have $\|\vec{X}_{i,:} - \widetilde{X}_{i,:}\|_2^2 \leq \epsilon_{\mathrm{sk}}/n$. Finally, let $M^* = U\Sigma V^\top$.*

*Then, we have*

- $\|\widehat{X} - M^*Y\|_F^2 \leq \Delta_u^2$;

- *If $\Delta_u \leq 0.1\sigma_{\min}(M^*)$, then $\mathrm{dist}(U, X) \leq 8\Delta_u/\sigma_{\min}(M^*)$;*

- *If $\Delta_u \leq 0.1\sigma_{\min}(M^*)$, then $\rho(X) \leq 8\mu/\sigma_{\min}(M^*)$;*

*where $\mathrm{dist}(U, X) = \min_{Q \in O_{k \times k}} \|UQ - X\|$ where $O_{k \times k}$ is the set of all $k \times k$ orthogonal matrices.*

Let us interpret the above lemma and explain why it's crucial to our final convergence analysis. For simplicity, suppose the noise $N = 0$ and $M^* = X^*Y^\top$, since $Y$ has orthonormal columns, $M^*Y = X^*$ and the first part states that if we solve the regression approximately and clip rows with large norms, then the approximate clipped matrix $\widehat{X}$ is close to $X^*$. The next two parts state that as long as $\widehat{X}$ and $X^*$ are close enough, then two crucial properties are guaranteed: 1). the distance between the space spanned by left singular vectors and $X$, the QR factor of the clipped matrix $\overline{X}$, is small and 2). the generalized incoherence of $X$ is small. These guarantees lead to a natural inductive argument: suppose $\Delta_u$ is small enough, then by our algorithm, we know that $\widehat{X}$ and $M^*Y$ are close and consequently $\mathrm{dist}(X, U)$ and $\rho(\widehat{X})$ are small. These two conditions serve as a basis to prove that for the next iteration, we still have $\widehat{X}$ and $M^*Y$ is small enough and the induction can proceed.

We want to highlight the major challenges in proving these assertions. Note that the induction argument effectively provides bounds on both subspace distance and generalized incoherence, and both notions heavily rely on the conditioning of intermediate matrices. The original analysis of Li et al. (2016) gives quantitative bounds on condition numbers assuming the updates are computed exactly, but the picture becomes much less clear when the updates are only computed approximately. Nevertheless, we prove that when the approximate updates are close enough to the optimality, then these bounds still hold. To compute these updates to high-precision, we utilize the high-accuracy, weighted multiple response solver being developed. One could view our proof as a mixture of algorithm and analysis: our analysis mandates the algorithm to provide strong guarantees, and we in turn design algorithms to achieve these goals. For more details, we refer readers to Appendix D.

## 4.3 MAIN RESULT

Our main theorem is as follows:

**Theorem 4.6** (Formal version of Theorem 1.1). *Given a noisy, possibly higher-rank observation $M \in \mathbb{R}^{n \times n}$ where $M^*$ is the rank-$k$ ground truth and $N$ is the noise matrix that satisfies Assumption 2.3. There is an algorithm (Algorithm 1) uses random initialization, runs in $O(\log(1/\epsilon))$ iterations and generates an $n \times n$ matrix $\widetilde{M}$ such that*

$$\|\widetilde{M} - M^*\| \leq O(\alpha^{-1}k\tau)\|W \circ N\| + \epsilon,$$

*The total running time is*

$$\widetilde{O}((\|W\|_0 \cdot k + nk^3)\log(1/\epsilon)).$$

Due to space limitation, we delay the proof of Theorem 4.6 to Appendix J. We want to briefly remark that our algorithm can be easily extended to cases where both $W$ and $M$ are rectangular matrices of size $m \times n$ as none of our analyses rely on the matrix being square. One could replace the factor $n$ in our runtime by $\max\{m, n\}$ when dealing with rectangular weighted low rank approximation.

## 4.4 COMPARISONS WITH RECENT WORKS

In this section, we provide a brief overview and comparison with other recent works, which could be classified into 3 categories: 1). slower, exact alternating minimization for weighted low rank approximation (Li et al., 2016); 2). faster, approximate alternating minimization for noiseless low rank matrix completion, a strictly simpler problem (Gu et al., 2024b) and 3). new metrics for measuring the effectiveness of low rank matrix factorizations (Yalcin et al., 2022; Zhang et al., 2024).

Compared to the result of Li et al. (2016), we significantly improve the running time from $O((\|W\|_0 k^2 + nk^3)\log(1/\epsilon))$ to $\widetilde{O}((\|W\|_0 k + nk^3)\log(1/\epsilon))$. For moderately large $k$ (say

$k = O(\sqrt{n})$) and dense weight matrix (say $\|W\|_0 = O(n^2)$), the Li et al. (2016) algorithm would take $O(n^3 \log(1/\epsilon))$ time, while ours only takes $\widetilde{O}(n^{2.5} \log(1/\epsilon))$ time. In the noisy matrix completion setting Kelner et al. (2023), $W$ is a Boolean matrix with $\|W\|_0 = \widetilde{O}(nk^{2+o(1)})$, applying our algorithm leads to an overall runtime of $\widetilde{O}(nk^{3+o(1)} \log(1/\epsilon))$, nearly matches the state-of-the-art (Kelner et al., 2023) In contrast, the Li et al. (2016) algorithm has a runtime of $\widetilde{O}(nk^{4+o(1)} \log(1/\epsilon))$. Moreover, our analysis accounts for the approximated computation at each step, thus it opens up the gate for further speedup. This also better depicts the picture of *practical* alternating minimization algorithms, where updates are computed approximately both due to floating point errors and efficiency concerns. We believe this also lays a foundation for theoretically verifying why alternating minimization with approximate updates has great empirical success.

Compared to the result of Gu et al. (2024b), we note they show for the simpler problem of noiseless matrix completion, the alternating minimization procedure could be sped up and run in $\widetilde{O}(\|W\|_0 \cdot k \log(1/\epsilon))$ time. Our result, even in the matrix completion setting, is a strict generalization of theirs, as they assume access to the entries of the ground truth $M^*$. In contrast, our model can only access noisy entries $M = M^* + N$, thus our recovery result suffers an error in the form of $O(k\tau) \cdot \|W \circ N\|$, which is 0 if $N = \mathbf{0}_{n \times n}$. We also provide spectral norm error guarantee rather than Frobenius norm error, which is the objective Gu et al. (2024b) obtains. The spectral norm is oftentimes considered more robust than the Frobenius norm. In terms of analysis, the proof approach of Gu et al. (2024b) is particularly geared towards noiseless matrix completion, while our analysis is much more general, as it can account for noisy matrix completion and weighted low rank approximation. We believe the generality and simplicity of our framework could be further extended to analyze alternating minimization for other problems, such as robust PCA and multi-view learning.

Compared to the results of Yalcin et al. (2022); Zhang et al. (2024), we focus on providing theoretical guarantees on the algorithm's performance, whereas Yalcin et al. (2022); Zhang et al. (2024) focus more on analyzing the optimization landscape and proposing complexity metrics for low rank matrix problems than developing specific algorithms for weighted low rank approximation. Specifically, the main contribution of Zhang et al. (2024) is developing a new complexity metric to characterize the difficulty of the nonconvex landscape arising from the Burer-Monteiro factorization. This metric aims to quantify when local search methods can successfully solve the factorized problem. The main contribution of Yalcin et al. (2022) is constructing a class of low-complexity matrix completion problem instances that can be solved in polynomial time, but for which the popular Burer-Monteiro factorization approach fails. Yalcin et al. (2022) also shows the existence of problem instances in this class that have exponentially many spurious local minima when using the Burer-Monteiro factorization, even though the original problem has a unique global solution. It would be interesting to study whether alternating minimization could also provide provable guarantees against these metrics and problems and in turn be accelerated.

## 5    CONCLUSION

In this paper, we study the weighted low rank approximation problem and efficient algorithm to solve it under mild recovery assumptions. Alternating minimization has been shown to be a powerful algorithmic prototype for this problem (Li et al., 2016), and we provide a fast, approximate implementation together with a robust error analysis for the framework. To this end, we improve the running time of Li et al. (2016) from $O((\|W\|_0 k^2 + nk^3) \log(1/\epsilon))$ to $\widetilde{O}((\|W\|_0 k + nk^3) \log(1/\epsilon))$. Our error analysis also serves as a theoretical explanation of why alternating minimization works well in practice especially when these updates are computed approximately for better efficiency.

We would also like to point out that the runtime of our algorithm is nearly linear in terms of solution verification. Given the weight matrix $W$ and a pair of low rank factors $X$ and $Y$, it takes $O(k)$ to verify a single entry of $W \circ (XY^\top)$ and we would need to verify a total of $\|W\|_0$ entries. However, it is also worth noting that such runtime can only be achieved when random initialization is used as if one resorts to SVD initialization, the initialization time becomes $O(n^3)$ which would dominate the overall runtime. It will be an interesting open problem whether we can further speed up the initialization using procedures such as random SVD and obtain a nearly linear time algorithm for alternating minimization with SVD initialization.

ACKNOWLEDGEMENT

This work was mostly done when Junze Yin was at Boston University. Junze Yin is supported by the Rice University graduate fellowship. Lichen Zhang is supported in part by NSF CCF-1955217 and NSF DMS-2022448.

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

APPENDIX

**Roadmap.** In Section A, we provide several basic definitions and tools. In Section B, we discuss more related work. In Section C, we describe the fast multiple response regression solver used to speed up the alternating minimization step. In Section D, we provide our key lemmas for the update step. In Section E, we prove our induction lemma. In Section F, we state several tools from previous work. In Section G, we analyze the SVD initialization and present our main result. In Section H, we present the random initialization algorithm and analyze its properties. In Section I, we show how to prove the final guarantee of our main Theorem. In Section J, we present the complete proof of our main theorem.

## A BASIC DEFINITIONS AND ALGEBRA TOOLS

In Section A.1, we present the properties of the weight matrix. Moreover, we explain the algebra tools which are used for later proofs. In Section A.2, we present some basic algebraic inequalities. In Section A.3, we state a few simple facts about the norm properties.

### A.1 PROPERTIES OF WEIGHT MATRIX

Here, we present the properties of weighted matrices.

**Definition A.1.** For weight matrix $W$, we define

$$\|W\|_{\infty,1} := \max\{\max_{i \in [n]} \|W_{i,:}\|_1, \max_{j \in [n]} \|W_{:,j}\|_1\}$$

and

$$\|W\|_{\infty,2} := \max\{\max_{i \in [n]} \|W_{i,:}\|_2, \max_{j \in [n]} \|W_{:,j}\|_2\}.$$

**Lemma A.2.** Let $\gamma > 0$, if $\|W - \mathbf{1}_n \mathbf{1}_n^\top\| \leq \gamma n$, then we have

- *Part 1.* $\|W - \mathbf{1}_n \mathbf{1}_n^\top\|_F \leq n^{1.5}\gamma$

    - *Further* $\|W\|_F \leq n^{1.5}\gamma + n$

- *Part 2.* $\|W - \mathbf{1}_n \mathbf{1}_n^\top\|_{\infty,1} \leq n^{1.5}\gamma$

    - *Further* $\|W\|_{\infty,1} \leq n^{1.5}\gamma + n$

*Proof.* **Proof of Part 1.** We have

$$\begin{aligned}
\|W - \mathbf{1}_n \mathbf{1}_n^\top\|_F^2 &\leq n\|W - \mathbf{1}_n \mathbf{1}_n^\top\|^2 \\
&\leq n \cdot (\gamma n)^2 \\
&\leq n^3 \gamma^2,
\end{aligned} \tag{2}$$

where the first step follows from **Part 4** of Fact A.7, the second step follows from the assumption from the lemma statement, and the last step follows from simple algebra.

Moreover, by the triangle inequality, we have

$$\begin{aligned}
\|W\|_F &= \|W - \mathbf{1}_n \mathbf{1}_n^\top + \mathbf{1}_n \mathbf{1}_n^\top\|_F \\
&\leq \|W - \mathbf{1}_n \mathbf{1}_n^\top\|_F + \|\mathbf{1}_n \mathbf{1}_n^\top\|_F \\
&\leq n^{1.5}\gamma + \|\mathbf{1}_n \mathbf{1}_n^\top\|_F \\
&= n^{1.5}\gamma + n,
\end{aligned}$$

where the first step follows from simple algebra, the second step follows from the triangle inequality, the third step follows from Eq. (2), and the last step follows from the definition of the Frobenius norm.

**Proof of Part 2.** Note that

$$\|W - \mathbf{1}_n\mathbf{1}_n^\top\|_{\infty,1} \le \sqrt{n} \cdot \|W - \mathbf{1}_n\mathbf{1}_n^\top\|_{\infty,2}$$
$$\le \sqrt{n} \cdot \|W - \mathbf{1}_n\mathbf{1}_n^\top\|$$
$$\le n^{1.5}\gamma$$

By the triangle inequality, we have

$$\|W\|_{\infty,1} \le \|W - \mathbf{1}_n\mathbf{1}_n^\top\|_{\infty,1} + \|\mathbf{1}_n\mathbf{1}_n^\top\|_{\infty,1}$$
$$\le n^{1.5}\gamma + n.$$

$\square$

**Lemma A.3.** *Bounds on $\gamma$ lead to bounds on $\|W\|_{\infty,1}$. Specifically,*

- *Part 1. If $\gamma < 1/(10n^{1/6})$, then we have*

$$\gamma \cdot (\|W\|_{\infty,1}/n)^{1/2} < 1$$

- *Part 2. If $\gamma < 1/(10n^{1/2})$, then we have*

$$\gamma \cdot (\|W\|_{\infty,1})^{1/2} < 1$$

**Remark A.4.** In previous work (Li et al., 2016), they wrote the final bound as $\gamma < f/(\|W\|_{\infty,1}/n)^{1/2}$ where $f$ are factors not depending on $\gamma$. For example, $f = \text{poly}(\alpha^{-1}, k, \tau, \mu)$. Their bound technically is not complete, because $\|W\|_{\infty,1}$ is also function of $\gamma$. So, in our work, our Lemma A.3 further calculates the actual condition required by Li et al. (2016) and hence completes their correctness proof.

*Proof.* **Proof of Part 1.** We need that

$$\gamma \cdot (\|W\|_{\infty,1}/n)^{1/2} < 1$$

It suffices to show that

$$\gamma \cdot ((n^{1.5}\gamma + n)/n)^{1/2} < 1$$

The above equation is equivalent to

$$\gamma \cdot (n^{0.5}\gamma + 1)^{1/2} < 1$$

It is sufficient to show that

$$\gamma^{1.5}n^{0.25} + \gamma < 1$$

Thus, as long as

$$\gamma < 1/(10n^{1/6})$$

the promised bound is held.

**Proof of Part 2.** We need that

$$\gamma \cdot (\|W\|_{\infty,1})^{1/2} < 1$$

It suffices to show that

$$\gamma \cdot (n^{1.5}\gamma + n)^{1/2} < 1$$

It is sufficient to show that

$$\gamma^{1.5}n^{0.75} + \gamma n^{0.5} < 1$$

Thus, as long as

$$\gamma < 1/(10n^{1/2})$$

we have the desired result.

$\square$

## A.2 BASIC ALGEBRAIC INEQUALITIES

In this section, we introduce some basic inequalities.

**Fact A.5.** For any $x, y$ and $\epsilon \in (0, 1)$, we have
$$(x + y)^2 \geq (1 - \epsilon)x^2 - \epsilon^{-1}y^2$$

*Proof.* It suffices to show
$$x^2 + 2xy + y^2 \geq (1 - \epsilon)x^2 - \epsilon^{-1}y^2.$$
Re-organizing the above terms, we have
$$\epsilon x^2 + 2xy + (1 + \epsilon^{-1})y^2 \geq 0$$
Thus it suffices to show that
$$\epsilon x^2 + 2xy + \epsilon^{-1}y^2 \geq 0.$$
It is obvious that
$$\epsilon x^2 + \epsilon^{-1}y^2 \geq 2|xy|.$$
Thus, we can complete the proof. $\qquad\square$

**Fact A.6.** Let $n$ be an arbitrary positive integer. Let $a_i \geq 0$ and $b_i \geq 0$ for all $i \in [n]$. Then, the following two inequalities hold
$$\min_{i \in [n]}\{a_i\} \sum_{i \in [n]} b_i \leq \sum_{i \in [n]} a_i b_i \leq \max_{i \in [n]}\{a_i\} \sum_{i \in [n]} b_i$$
$$\min_{i \in [n]}\{b_i\} \sum_{i \in [n]} a_i \leq \sum_{i \in [n]} a_i b_i \leq \max_{i \in [n]}\{b_i\} \sum_{i \in [n]} a_i.$$

## A.3 PROPERTIES OF NORMS

We state some standard facts about norms without providing proofs.

**Fact A.7.** We have the following facts about norms:

- Part 1. For any matrix $A \in \mathbb{R}^{n \times n}$, let $A_j$ denote the $j$-th column of $A$. Then we have $\sum_{j=1}^n \|A_j\|_2^2 = \|A\|_F^2$.

- Part 2. For any psd matrix $A$, for any vector $x$, $x^\top A x \geq \sigma_{\min}(A)$.

- Part 3. Let $U \in \mathbb{R}^{n \times k}$ denote an orthonormal basis. Then for any $k \times k$ matrix $B$, we have $\|UB\| = \|B\|$.

- Part 4. For any matrix $A \in \mathbb{R}^{n \times k}$, we have $\|A\| \leq \|A\|_F \leq \sqrt{k}\|A\|$.

- Part 5. For any matrix $A$ and $B$, $\sigma_{\min}(A) \geq \sigma_{\min}(B) - \|A - B\|$

- Part 6. For any matrix $A \in \mathbb{R}^{n \times k}$ and any orthonormal basis $Q \in \mathbb{R}^{k \times k}$. $\sigma_{\min}(A) = \sigma_{\min}(AQ)$.

- Part 7. For any vector $x \in \mathbb{R}^k$ and for any orthornomal basis $Q \in \mathbb{R}^{k \times k}$, we have $\|x\|_2 = \|Qx\|_2$.

## A.4 GENERALIZED MATRIX INCOHERENCE

In this section, we provide a generalized notion of matrix incoherence, denoted by $\rho$.

**Definition A.8.** Let $A \in \mathbb{R}^{n \times k}$. The generalized incoherence of $A$ is denoted as $\rho(A)$, i.e.,
$$\rho(A) := \frac{n}{k} \cdot \max_{i \in [n]}\{\|A_{i,:}\|_2^2\}. \tag{3}$$

**Claim A.9.** *When $A \in \mathbb{R}^{n \times k}$ has orthonormal columns, $1 \leq \rho(A) \leq \frac{n}{k}$.*

*Proof.* Since $A$ is an orthogonal matrix, $\|A_{i,:}\|_2^2 \leq 1$ for all $i \in [n]$, and thus $\rho(A) \leq \frac{n}{k}$. In addition, $\sum_{i=1}^n \|A_{i,:}\|_2^2 = k$, we have $\max_{i \in [n]}\{\|A_{i,:}\|_2^2\} \geq \frac{k}{n}$ and then $\rho(A) \geq 1$. $\qquad\square$

## A.5 Angles and Distances Between Subspaces

An important metric we use in this paper to quantify the progress of our algorithm is the distance between subspaces. We illustrate these definitions below.

**Definition A.10.** Let $X, Y$ be $n \times k$ matrices with orthonormal columns, i.e., $X^\top X = I_k$ and $Y^\top Y = I_k$.

We define $\tan \theta(Y, X)$ to be equal to

$$\|Y_\perp^\top X (Y^\top X)^{-1}\|.$$

We define $\cos \theta(Y, X)$ to be equal to

$$\sigma_{\min}(Y^\top X);$$

we define $\sin \theta(Y, X)$ to be equal to

$$\|(I - YY^\top)X\|;$$

Let $O_k$ be a set containing all $k \times k$ orthogonal matrices. We define $\mathrm{dist}(Y, X)$ to be equal to

$$\min_{Q \in O_k} \|YQ - X\|.$$

Note that by their definitions, we can get

- $\cos \theta(Y, X) = 1/\|(Y^\top X)^{-1}\|$,
- $\cos \theta(Y, X) \leq 1$,
- $\sin \theta(Y, X) = \|Y_\perp Y_\perp^\top X\| = \|Y_\perp^\top X\|$, and
- $\sin \theta(Y, X) \leq 1$.

**Lemma A.11** (Structural lemma for orthonormal columns, Lemma A.5 of Gu et al. (2024b)). *We let $X$ and $Y$ to be arbitrary matrices in $\mathbb{R}^{n \times k}$ and both are orthogonal. Then, we can get*

$$(Y^\top X)_\perp = Y_\perp^\top X.$$

**Lemma A.12** (Lemma A.7 of Gu et al. (2024b)). *We let $X$ and $Y$ be two matrices in $\mathbb{R}^{n \times k}$ and both have orthonormal columns. Then, we have*

$$\tan \theta(Y, X) = \frac{\sin \theta(Y, X)}{\cos \theta(Y, X)}.$$

**Lemma A.13** (Lemma A.8 of Gu et al. (2024b)). *Let $X, Y \in \mathbb{R}^{n \times k}$ be orthogonal matrices. Then, we can get*

$$\sin^2 \theta(Y, X) + \cos^2 \theta(Y, X) = 1.$$

**Lemma A.14** (Lemma A.9 of Gu et al. (2024b)). *Let $X$ and $V$ be two matrices in $\mathbb{R}^{n \times k}$ with orthonormal columns, then, we can get*

- $\tan \theta(Y, X) \geq \sin \theta(Y, X)$
- $\tan \theta(Y, X) \geq \frac{1 - \cos \theta(Y, X)}{\cos \theta(Y, X)}$
- $\mathrm{dist}(Y, X) \geq \sin \theta(Y, X)$
- $\sin \theta(Y, X) + \frac{1 - \cos \theta(Y, X)}{\cos \theta(Y, X)} \geq \mathrm{dist}(Y, X)$
- $2 \tan \theta(Y, X) \geq \mathrm{dist}(Y, X)$

## B   MORE RELATED WORK

**Sketching**   To achieve the crucial speedup, we utilize sketching-based preconditioners and we therefore provide an overview of the sketching literature. Roughly speaking, given a tall dense matrix $A$, the goal of sketching is to design a family of random matrices $\Pi$ such that, if we randomly sample $S \sim \Pi$, we have

- $S$ has much smaller number of rows than $A$ (thus the matrix $SA$ is close to a square matrix rather than rectangular);
- $SA$ preserves singular values of $A$ with high probability;
- $S$ can be quickly applied to $A$.

Given such a family $\Pi$, it is natural to apply an $S$ to $A$ then solve the smaller problem directly. This is the so-called *sketch-and-solve* paradigm. Sketch-and-solve has led to the development of fast algorithms for many problems, such as linear regression (Clarkson & Woodruff, 2013; Nelson & Nguyên, 2013; Song et al., 2023f;d), linear and kernel SVMs (Gu et al., 2025), low rank approximation with Frobenious norm (Clarkson & Woodruff, 2013; Nelson & Nguyên, 2013), matrix CUR decomposition (Boutsidis & Woodruff, 2014; Song et al., 2017; 2019c), weighted low rank approximation (Razenshteyn et al., 2016), entrywise $\ell_1$ norm low rank approximation (Song et al., 2017; 2019b), tensor regression (Song et al., 2021a; Reddy et al., 2022; Diao et al., 2018; 2019), tensor low rank approximation (Song et al., 2019c), tensor power method (Deng et al., 2023b), and general norm column subset selection (Song et al., 2019a).

As modern machine learning centers around algorithms that are iterative in nature. Sketching can also be adapted to an iterative process to reduce the cost of iteration. This is the so-called *Iterate-and-sketch* approach and it has led to fast algorithms for many fundamental problems, such as linear programming (Cohen et al., 2021; Song & Yu, 2021; Jiang et al., 2021), empirical risk minimization (Lee et al., 2019; Qin et al., 2023), semi-definite programming (Gu & Song, 2022; Song et al., 2023e), John Ellipsoid computation (Song et al., 2022c), Frank-Wolfe algorithm (Xu et al., 2021; Song et al., 2022a), hamming estimation (Hu et al., 2024), reinforcement learning (Shrivastava et al., 2023), $k$ means clustering (Liang et al., 2022), online weighted matching problem (Song et al., 2025), barrier functions (Gu et al., 2024a), softmax-inspired regression (Deng et al., 2023a; Gao et al., 2025; Li et al., 2023b; Sinha et al., 2023; Li et al., 2023a; Song et al., 2023a; 2024), leverage score inspired regression (Li et al., 2024), federated learning (Song et al., 2023b; Bian et al., 2023), discrepancy problem (Deng et al., 2022; Song et al., 2022b), non-convex optimization (Song et al., 2021b;c; Alman et al., 2023; Zhang, 2022), and attention approximation (Gao et al., 2023a;b).

**(Weighted) low rank approximation**   Low rank approximation has emerged as a crucial technique in machine learning and numerical linear algebra, enabling the extraction of essential structures from high-dimensional data while reducing computational costs. The goal is to find $X, Y \in \mathbb{R}^{n \times k}$ which minimizes $\|M - XY^\top\|_F$. It has been applied to numerous fields, including training deep neural networks (Song et al., 2021c), approximating attention mechanisms (Alman & Song, 2023; 2024; Chen et al., 2024), maintaining dynamic Kronecker products (Song et al., 2023c), and tensor product regression (Reddy et al., 2022). In many practical scenarios, certain entries of $M$ hold greater significance than others, giving rise to weighted low-rank approximation, where the objective is to minimize $\|W \circ (M - XY^\top)\|_F$ for some weight matrix $W \in \mathbb{R}_{\geq 0}^{n \times n}$ (Li et al., 2016; Razenshteyn et al., 2016; Gu et al., 2024b; Liang et al., 2024).

## C   WEIGHTED MULTIPLE RESPONSE REGRESSION SOLVERS

In this section, we show how to solve weighted multiple response regression by solving standard linear regressions. We present randomized and fast regression solvers based on sketching and preconditioning.

### C.1   GENERIC REDUCTION AND ERROR CONVERSION

In this section, we present a generic framework to reduce the weighted multiple response regression problem to solving $O(n)$ ordinary least-square regressions. This simple and efficient reduction enables

us to deploy fast regression solvers to handle the approximate updates in alternating minimization. We also present a tool that converts the relative error on the regression cost to the quality of approximate solution.

The first lemma states that the cost of a weighted multiple response regression can be decomposed into a summation of $n$ weighted linear regressions.

**Claim C.1.** *Given matrices $M, W \in \mathbb{R}^{n \times n}$ and $X, Y \in \mathbb{R}^{n \times k}$, we have*

$$\min_{X \in \mathbb{R}^{n \times k}} \|M - XY^\top\|_W^2 = \sum_{i=1}^{n} \min_{X_{i,:} \in \mathbb{R}^k} \|D_{\sqrt{W_i}} Y X_{i,:} - D_{\sqrt{W_i}} M_{i,:}\|_2^2,$$

*and*

$$\min_{Y \in \mathbb{R}^{n \times k}} \|M - XY^\top\|_W^2 = \sum_{i=1}^{n} \min_{Y_{i,:} \in \mathbb{R}^k} \|D_{\sqrt{W_i}} X Y_{i,:} - D_{\sqrt{W_i}} M_{:,i}\|_2^2.$$

*Proof.* Since the two equations can be proved in a similar way, we only prove the first one.

$$
\begin{aligned}
\min_{X \in \mathbb{R}^{n \times k}} \|M - XY^\top\|_W^2 &= \min_{X \in \mathbb{R}^{n \times k}} \sum_{i,j} W_{i,j} (XY^\top - M)_{i,j}^2 \\
&= \min_{X \in \mathbb{R}^{n \times k}} \sum_{i=1}^{n} \|D_{\sqrt{W_i}} (Y X_{i,:} - M_{i,:})\|_2^2 \\
&= \min_{X \in \mathbb{R}^{n \times k}} \sum_{i=1}^{n} \|D_{\sqrt{W_i}} Y X_{i,:} - D_{\sqrt{W_i}} M_{i,:}\|_2^2 \\
&= \sum_{i=1}^{n} \min_{X_{i,:} \in \mathbb{R}^k} \|D_{\sqrt{W_i}} Y X_{i,:} - D_{\sqrt{W_i}} M_{i,:}\|_2^2,
\end{aligned}
$$

where the 1st step is due to $\|A\|_W^2$'s definition, the 2nd step is by rewriting each row as an independent regression problem, the 3rd step follows from simple algebra, and the last step follows from the fact that there is no $X$ in $\min_{X \in \mathbb{R}^{n \times k}} \sum_{i=1}^{n} \|D_{\sqrt{W_i}} Y X_{i,:} - D_{\sqrt{W_i}} M_{i,:}\|_2^2$ but only $X_{i,:}$. Thus, we complete the proof. $\square$

The next lemma provides a simple conversion of weighted linear regression to ordinary least-squares, via a scaling trick.

**Lemma C.2** (Lemma B.6 of Gu et al. (2024b)). *Let $A$ be a real $n \times d$ matrix with $n \geq d$, $b$ be an $n$-dimensional real vector and $w$ be a non-negative $n$-dimensional vector (weight). Let $\epsilon_0 \in (0, 0.1)$ be accuracy parameter and $\delta_0 \in (0, 0.1)$ controls failure probability. Suppose that $\mathcal{T}(n, d, \epsilon_0, \delta_0)$ is the running time of a regression solver, and $x' \in \mathbb{R}^d$ is the output of the regression solver satisfying*

$$\|Ax' - b\|_2 \leq (1 + \epsilon_0) \min_{x \in \mathbb{R}^d} \|Ax - b\|_2$$

*with probability at least $1 - \delta_0$.*

*Then, there exists an algorithm whose running time is*

$$O(\mathrm{nnz}(A)) + \mathcal{T}(n, d, \epsilon_0, \delta_0)$$

*and outputs a vector $x' \in \mathbb{R}^d$, which satisfy*

$$\|Ax' - b\|_w \leq (1 + \epsilon_0) \min_{x \in \mathbb{R}^d} \|Ax - b\|_w$$

*with probability at least $1 - \delta_0$.*

One of the main reasons Li et al. (2016) resorts to exact weighted multiple response regression is that most approximate solvers provide backward error guarantees on the cost of regression. On the other hand, we would like the *approximate solution* of the regression to be close to the exact solution. The following lemma converts the backward error on the cost, to the forward error on the solution.

**Lemma C.3** (Backward error, Lemma B.5 in Gu et al. (2024b)). *Let $A$ be a real $n \times d$ matrix with $n \geq d$, $b$ be an $n$-dimensional real vector. Let $x_{\mathrm{OPT}}$ be the exact solution to the regression problem*

$$\min_x \|Ax - b\|_2.$$

*Suppose that there exists a vector $x' \in \mathbb{R}^d$, satisfying*

$$\|Ax' - b\|_2 \leq (1 + \epsilon) \min_{x \in \mathbb{R}^d} \|Ax - b\|_2.$$

*Then, we have*

$$\|x' - x_{\mathrm{OPT}}\|_2 \leq O(\sqrt{\epsilon}) \cdot \frac{1}{\sigma_{\min}(A)} \cdot \|Ax_{\mathrm{OPT}} - b\|_2.$$

Before wrapping up this section, we present a meta algorithm for solving weighted multiple response regression.

---

**Algorithm 3** Fast, high precision solver for weighted multiple response regression

---

1: **procedure** MULTIPLEREGRESSION($A \in \mathbb{R}^{n \times n}, B \in \mathbb{R}^{n \times k}, W \in \mathbb{R}^{n \times n}$)
2:                                                            ▷ $A_i$ is the $i$-th column of $A$
3:                                                             ▷ $W_i$ is the $i$-th column of $W$
4:            ▷ $D_{W_i}$ is a diagonal matrix where we put $W_i$ on diagonal, other locations are zero
5:       $X_i \leftarrow \min_{x \in \mathbb{R}^k} \|D_{W_i} Bx - D_{W_i} A_i\|_2$
6:       **return** $X$                                                      ▷ $X \in \mathbb{R}^{k \times n}$
7: **end procedure**
8:
9: **procedure** FASTMULTIPLEREGRESSION($A \in \mathbb{R}^{n \times n}, B \in \mathbb{R}^{n \times k}, W \in \mathbb{R}^{n \times n}$)
10:                                                     ▷ $A_i$ is the $i$-th column of $A$
11:                                                  ▷ $W_i$ is the $i$-th column of $W$
12:            ▷ $D_{W_i}$ is a diagonal matrix where we put $W_i$ on diagonal, other locations are zero
13:       $X_i \leftarrow$ HIGHPRECISIONREG($D_{W_i} B, D_{W_i} A_i, \epsilon, \delta$)                  ▷ Algorithm 2
14:       **return** $X$                                             ▷ $X \in \mathbb{R}^{k \times n}$
15: **end procedure**

---

## C.2 LOW ACCURACY SOLVER

We provide an algorithm that uses a sparse sketching matrix to obtain a low accuracy solution (inverse polynomial dependence on accuracy parameter $\epsilon$).

**Definition C.4** (OSNAP matrix, (Nelson & Nguyên, 2013)). For every sparsity parameter $s$, target dimension $m$, and positive integer $d$, the OSNAP matrix with sparsity $s$ is defined as

$$S_{r,j} = \frac{1}{\sqrt{s}} \cdot \delta_{r,j} \cdot \sigma_{r,j},$$

for all $r \in [m], j \in [d]$, where $\sigma_{r,j}$ are independent Rademacher random variables and $\delta_{r,j}$ are Bernoulli random variables with

- For every $i \in [d]$, $\sum_{r \in [m]} \delta_{r,i} = s$, which means each column of $S$ contains exactly $s$ nonzero entries.

- For all $r \in [m]$ and $i \in [d]$, $\mathbb{E}[\delta_{r,i}] = s/m$.

- $\forall T \in [m] \times [d]$, $\mathbb{E}[\prod_{(r,i) \in T} \delta_{r,i}] \leq \prod_{(r,i) \in T} \mathbb{E}[\delta_{r,i}] = (s/m)^{|T|}$, i.e., $\delta_{r,i}$ are negatively correlated.

Crucially, the OSNAP matrix produces a subspace embedding with nearly linear in $d$ row count.

**Lemma C.5** ((Cohen, 2016)). *Let $S \in \mathbb{R}^{m \times n}$ be an OSNAP matrix as in Def. C.4.*

*Let $\epsilon, \delta \in (0,1)$ be parameters.*

*For any integer $d \leq n$, if*

- *$m = O(\epsilon^{-2} d \log(d/\delta))$;*

- *$s = O(\epsilon^{-1} \log(d/\delta))$,*

*then an $s$-sparse OSNAP matrix $S$ is an $(\epsilon, \delta)$ oblivious subspace embedding, i.e., for any fixed orthonormal basis $U \in \mathbb{R}^{n \times d}$ with probability at least $1 - \delta$, and the singular values of $SU$ lie in $[1 - \epsilon, 1 + \epsilon]$.*

To distinguish with $\epsilon, \delta$ for our final algorithm, here we use $\epsilon_0, \delta_0$ for the subroutine (approximate linear regression).

**Lemma C.6** (Input sparsity and low accuracy regression). *Given a matrix $A \in \mathbb{R}^{n \times d}$ and a vector $b \in \mathbb{R}^n$, let $\epsilon_0 \in (0, 0.1)$ and $\delta_0 \in (0, 0.1)$, there exists an algorithm that takes time*

$$O((\epsilon_0^{-1} \operatorname{nnz}(A) + \epsilon_0^{-2} d^3) \cdot \log(d/\delta_0))$$

*and outputs $x' \in \mathbb{R}^d$ such that*

$$\|Ax' - b\|_2 \leq (1 + \epsilon_0) \min_{x \in \mathbb{R}^d} \|Ax - b\|_2$$

*holds with probability $1 - \delta_0$.*

*Proof.* To obtain desired accuracy and probability guarantee, we pick $S$ to be an OSNAP (Definition C.4) with

$$m = O(\epsilon_0^{-2} d \log(d/\delta_0))$$

and

$$s = O(\epsilon_0^{-1} \log(d/\delta_0)).$$

We simply apply $S$ to $A$ then solve the sketched regression $\min_{x \in \mathbb{R}^d} \|SAx - Sb\|_2$.

- As $S$ is a matrix where each column only has $s$ nonzero entries, the time to compute $SA$ is

$$O(s \operatorname{nnz}(A)) = O(\epsilon_0^{-1} \operatorname{nnz}(A) \log(d/\delta_0)).$$

- The regression can then be solved via normal equation, i.e.,

$$(A^\top S^\top S A)^\dagger A^\top S^\top b.$$

The time to form the Gram matrix is

$$O(md^2),$$

computing the $d \times d$ inversion takes $O(d^3)$ time, and forming the final solution takes another

$$O(md^2)$$

time. Overall, this gives a runtime of

$$O(\epsilon_0^{-2} d^3 \log(d/\delta_0)).$$

Thus, the overall runtime is

$$O((\epsilon_0^{-1} \operatorname{nnz}(A) + \epsilon_0^{-2} d^3) \cdot \log(d/\delta_0)).$$

$\square$

## C.3 High Accuracy Solver

Our key algorithmic ingredient is a high accuracy, iterative and sketching-based solver for regression. The sketching matrix we will be using is the dense subsampled randomized Hadamard transform SRHT due to loss of structure in the iterative process.

**Definition C.7** (Subsampled randomized Hadamard transform (SRHT), (Lu et al., 2013))**.** Let $P$ be a random sampling matrix in $\{0,1\}^{m \times n}$ and for each row of $P$, there exists a 1 at a uniformly random position.

Let $H \in \{-1,1\}^{n \times n}$ be the Hadamard matrix.

Let $D \in \mathbb{R}^{n \times n}$ be a diagonal matrix, whose diagonal entries are all in $\{-1,+1\}$ with the same probability.

We define the SRHT matrix $S \in \mathbb{R}^{m \times n}$ as

$$S := \frac{1}{\sqrt{m}} PHD.$$

**Remark C.8.** For a real $n \times d$ matrix $A$ it takes $O(nd \log n)$ time to apply $S$ to $A$.

**Lemma C.9.** *Let $S \in \mathbb{R}^{m \times n}$ be an SRHT matrix (see Definition C.7), $\epsilon, \delta \in (0,1)$ be parameters. Let $d$ be an arbitrary integer, which is less than or equal to $n$. Suppose $m = O(\epsilon^{-2} d \log^2(n/\delta))$. Let $U \in \mathbb{R}^{n \times d}$ be a fixed orthonormal basis.*

*We say that $S$ is an $(\epsilon, \delta)$-oblivious subspace embedding if the singular values of the matrix $SU$ are in the interval $[1 - \epsilon, 1 + \epsilon]$, with probability at least $1 - \delta$.*

**Lemma C.10** (Dense and high accuracy regression, Lemma B.1 in Gu et al. (2024b))**.** *Let $A$ be a real $n \times d$ matrix, $b$ be a real $n$-dimensional vector, $\epsilon \in (0, 0.1)$ be an accuracy parameter and $\delta \in (0, 0.1)$ be the failure probability. Then, there exists an algorithm that takes time*

$$O((nd \log n + d^3 \log^2(n/\delta)) \log(1/\epsilon))$$

*and outputs a vector $x' \in \mathbb{R}^d$ satisfying*

$$\|Ax' - b\|_2 \leq (1 + \epsilon) \min_{x \in \mathbb{R}^d} \|Ax - b\|_2$$

*with probability at least $1 - \delta$.*

## D Key Property for Robust Update

In this section, we prove crucial properties of the algorithm that enable the approximate updates. In Section D.1, we formally define several necessary notations needed to analyze our robust updated step. In Section D.2, we analyze the key properties of our robust update step.

### D.1 Definitions for Update Step

We present a closed-form solution for linear regression via normal equation.

**Fact D.1.** Define $x := \arg\min_x \|Ax - b\|_2$, then we have

$$x = (A^\top A)^{-1} A^\top b.$$

Similarly, for weighted regression, we define

$$x := \arg\min_x \|D_{W_i} Ax - D_{W_i} b\|_2.$$

Then, we have

$$x = (A^\top D_{W_i} A)^{-1} A^\top D_{W_i} b$$

**Definition D.2.** We define $\xi$ as

$$\xi := \mu k/n.$$

$\xi$ captures the maximum incoherence of the ground truth.

**Definition D.3.** We define $\eta \geq 1$ to be parameter that distinguish random and SVD initialization.

- For random initialization, we set $\eta := \mu k$.

- For SVD initialization, we set $\eta := 1$.

We next define the value choice of $\gamma$, which controls how far away the weight matrix $W$ can be from the all-1's matrix.

**Definition D.4.** Let

$$\gamma \leq \frac{1}{100} \cdot \frac{\alpha}{\text{poly}(k, \tau, \mu) \cdot n^{c_0}}$$

where $c_0$ is a fixed constant between $(0, 1/2]$.

For the convenience of analysis, we define the following threshold parameters:

**Definition D.5.** Let $C \geq 10^5$ denote a sufficiently large constant. We define

$$\Delta_d := C\alpha^{-1.5}\mu^{1.5}k^2\gamma(\|W\|_{\infty,1}/n)^{1/2} + C\alpha^{-1}\eta\mu k^2\tau^{0.5}\gamma$$
$$\Delta_f := C\alpha^{-1}\eta k.$$

The choice of $\Delta_d$ is decided in Eq. (9) and Eq. (15). The choice of $\Delta_f$ is decided in Eq. (16).

The next two definitions capture the error gap of our algorithm.

**Definition D.6.** Let $\Delta_d$ and $\Delta_f$ be defined as Definition D.5. We define

$$\Delta_u := \Delta_d \cdot \text{dist}(Y, V) + \Delta_f \cdot \|W \circ N\|.$$

and

$$\Delta_g := 0.01\Delta_d \cdot \text{dist}(Y, V) + 0.01\Delta_f \cdot \|W \circ N\| + 2\sqrt{\epsilon_{\text{sk}}}$$

By properly controlling the error $\epsilon_{\text{sk}}$, we can show that $\Delta_g$ is a constant factor smaller than $\Delta_u$.

**Claim D.7.** *If the following condition holds*

$$\epsilon_{\text{sk}} \leq 10^{-4}\Delta_f^2 \cdot \|W \circ N\|^2,$$

*then we have*

$$\Delta_g \leq 0.1\Delta_u.$$

*Proof.* We have

$$\begin{aligned}
\Delta_g &= 0.01\Delta_d \cdot \text{dist}(Y, V) + 0.01\Delta_f \cdot \|W \circ N\| + 2\sqrt{\epsilon_{\text{sk}}} \\
&\leq 0.01\Delta_d \cdot \text{dist}(Y, V) + 0.01\Delta_f \cdot \|W \circ N\| + 0.02\Delta_f \cdot \|W \circ N\| \\
&\leq 0.1\Delta_u
\end{aligned}$$

where the second step follows from condition on $\epsilon_{\text{sk}}$, and the last step follows from the definition of $\Delta_u$. $\qquad\square$

By setting the error and failure probability appropriately, we can show the extra blowups in our algorithm are of the order $\text{poly} \log n$.

**Claim D.8.** *By the choice of $\epsilon_{\text{sk}}$, we have*

$$\log(n/\epsilon_{\text{sk}}) = O(\log(n))$$

*By choice of failure probability ($\delta_0$) of sketch,*

$$\log(n/\delta_0) = \log(n \log(1/\epsilon))$$

*Proof.* By assumption on $W$, we can see that

$$\|W\|_\infty \leq \text{poly}(n).$$

Since $W \circ N$ is a noisy part, it is also natural to consider

$$\|W \circ N\|_\infty \leq \text{poly}(n),$$

otherwise, it is not interesting.

Since the noise cannot be $0$, thus it is natural to assume that

$$\|N\|_\infty \geq 1/\text{poly}(n).$$

Thus, we know

$$1/\text{poly}(n) \leq \|W \circ N\|_F \leq \text{poly}(n). \tag{4}$$

We also know that $k \leq n$.

Now, we can compute

$$\begin{aligned}
\log(n/\epsilon_{\text{sk}}) &\leq O(\log(n/(\Delta_f^2 \|W \circ N\|_F^2))) \\
&\leq O(\log(n/\|W \circ N\|_F^2)) \\
&\leq O(\log(n)),
\end{aligned}$$

where the first step follows from we choose $\epsilon_{\text{sk}} = \Theta(\Delta_f^2 \|W \circ N\|_F^2)$, the second step follows from $\Delta_f \geq 1$, the third step follows from $\|W \circ N\|_F^2 \geq 1/\text{poly}(n)$ (see Eq. (4)).

Sum over all the $T = O(\log(1/\epsilon))$ iterations, so

$$\log(n/\delta_0) = O(\log(n \log(1/\epsilon))).$$

$\square$

## D.2 KEY LEMMA FOR ROBUST UPDATE STEP

Table 2: For convenience, we provide a table to summarize the notations in Lemma D.9.

| Notation | Meaning |
|---|---|
| $\widetilde{X}$ | Optimal matrix for exact regression |
| $\overline{X}$ | Clipped matrix of $\widetilde{X}$ |
| $\vec{X}$ | Optimal matrix for sketched regression |
| $\widehat{X}$ | Clipped matrix of $\vec{X}$ |

---

**Algorithm 4** Clipping rows whose norms are larger than a constant factor of $\xi$.

---

1: **procedure** CLIP($\widetilde{X} \in \mathbb{R}^{n \times k}$)
2:      $\xi \leftarrow \frac{\mu k}{n}$
3:      **for** $i = 1$ to $n$ **do**
4:          **if** $\|\widetilde{X}_{i,:}\|_2^2 \leq 4\xi$ **then**
5:              $\overline{X}_{i,:} \leftarrow \widetilde{X}_{i,:}$
6:          **else**
7:              $\overline{X}_{i,:} \leftarrow 0$
8:          **end if**
9:      **end for**
10:     **return** $\overline{X}$
11: **end procedure**

---

Here, we analyze the key properties of the robust update step.

**Lemma D.9** (Key lemma for update step). *Let $Y \in \mathbb{R}^{n \times k}$ be a (column) orthogonal matrix. Let $\xi$ be defined as Definition D.2. Let $\Delta_u$ be defined as Definition D.6.*

*We define matrix $\widetilde{X} \in \mathbb{R}^{n \times k}$ as follows:*

$$\widetilde{X} := \arg \min_{X \in \mathbb{R}^{n \times k}} \|M - XY^\top\|_W,$$

*We define matrix $\overline{X} \in \mathbb{R}^{n \times k}$ as follows*

$$\overline{X}_{i,:} := \begin{cases} \widetilde{X}_{i,:} & \text{if } \|\widetilde{X}_{i,:}\|^2 \leq 4\xi \\ 0 & \text{otherwise} \end{cases}$$

*We define $X \in \mathbb{R}^{n \times k}, R \in \mathbb{R}^{k \times k}$ to the QR decomposition of $\overline{X}$, i.e. $\overline{X} = XR$.*

*We define $\vec{X} \in \mathbb{R}^{n \times k}$ to be the sketch solution such that for all $i \in [n]$*

$$\|\vec{X}_{i,:} - \widetilde{X}_{i,:}\|^2 \leq \epsilon_{\mathrm{sk}}/n.$$

*We define $\widehat{X} \in \mathbb{R}^{n \times k}$ to denote the clip of the sketched solution. Recall that $M^* = U\Sigma V^\top$. Then,*

- *Part 1.*

$$\|\widehat{X} - U\Sigma V^\top Y\|_F^2 \leq \Delta_u^2;$$

- *Part 2. If $\Delta_u < 0.1\sigma_{\min}(M^*)$, then*

$$\mathrm{dist}(U, X) \leq 8\Delta_u/\sigma_{\min}(M^*)$$

- *Part 3. If $\Delta_u < 0.1\sigma_{\min}(M^*)$, then*

$$\rho(X) \leq 8\mu/\sigma_{\min}(M^*)^2$$

*Proof.* **Proof of Part 1.** Recall the weighted multiple response regression

$$\min_{X \in \mathbb{R}^{n \times k}} \|M - XY^\top\|_W^2,$$

The above problem can be written as $n$ different regression problems. The $i$-th linear regression has the formulation

$$\min_{X_{i,:} \in \mathbb{R}^k} \|D_{\sqrt{W_i}} Y X_{i,:} - D_{\sqrt{W_i}} M_{i,:}\|^2.$$

We have

$$\begin{aligned}
\widetilde{X}_{i,:}^\top &= M_{i,:}^\top \cdot D_{W_i} Y (Y^\top D_{W_i} Y)^{-1} \\
&= ((M^*)_{i,:}^\top + N_{i,:}^\top) \cdot D_{W_i} Y (Y^\top D_{W_i} Y)^{-1} \\
&= (M^*)_{i,:}^\top \cdot D_{W_i} Y (Y^\top D_{W_i} Y)^{-1} + N_{i,:}^\top \cdot D_{W_i} Y (Y^\top D_{W_i} Y)^{-1}, \tag{5}
\end{aligned}$$

where the first step follows from the Fact D.1, and the second step follows from $M_{i,:}^\top = (M^*)_{i,:}^\top + N_{i,:}^\top$ (because $M = M^* + N$), and the third step follows from simple algebra.

Given $M^* = U\Sigma V^\top$, the first term in Eq. (5) can be rewritten as follows:

$$\begin{aligned}
&(M^*)_{i,:}^\top \cdot D_{W_i} Y (Y^\top D_{W_i} Y)^{-1} \\
&= U_{i,:}^\top \cdot \Sigma V^\top D_{W_i} Y (Y^\top D_{W_i} Y)^{-1} \\
&= U_{i,:}^\top \cdot \Sigma V^\top (YY^\top + Y_\perp Y_\perp^\top) D_{W_i} Y (Y^\top D_{W_i} Y)^{-1} \\
&= U_{i,:}^\top \cdot \Sigma V^\top YY^\top D_{W_i} Y (Y^\top D_{W_i} Y)^{-1} + U_{i,:}^\top \Sigma V^\top Y_\perp Y_\perp^\top D_{W_i} Y (Y^\top D_{W_i} Y)^{-1} \\
&= U_{i,:}^\top \cdot \Sigma V^\top Y + U_{i,:}^\top \Sigma V^\top Y_\perp Y_\perp^\top D_{W_i} Y (Y^\top D_{W_i} Y)^{-1}, \tag{6}
\end{aligned}$$

where the first step follows from the fact that $(M^*)_{i,:}^\top = U_{i,:}^\top \Sigma V^\top$, the second step follows from $I = YY^\top + Y_\perp Y_\perp^\top$, the third step follows from simple algebra, and the last step follows from $AA^{-1} = I$.

Combining Eq. (5) and Eq. (6) we have

$$\widetilde{X}_{i,:}^\top - U_{i,:}^\top \Sigma V^\top Y = U_{i,:}^\top \Sigma V^\top Y_\perp Y_\perp^\top D_{W_i} Y (Y^\top D_{W_i} Y)^{-1} + N_{i,:}^\top D_{W_i} Y (Y^\top D_{W_i} Y)^{-1} \quad (7)$$

We define set $T \subset [n]$ as follows

$$T := \{i \in [n] \mid \sigma_{\min}(Y^\top D_{W_i} Y) \le 0.25\alpha/\eta\}. \quad (8)$$

We upper bound $|T|$ in different ways for SVD initialization and random initialization.

**SVD case.** We have $\eta = 1$. Using Lemma F.1 and choose $\epsilon = \Theta(1)$, we have

$$\begin{aligned}
|T| &\le 10^5 \cdot \alpha^{-3}\mu^2 k^3 \gamma^2 \cdot \|W\|_{\infty,1} \cdot \|V - Y\|^2 \\
&= 10^5 \cdot \alpha^{-3}\mu^2 k^3 \gamma^2 \cdot \mu k \cdot (\|W\|_{\infty,1}/n) \cdot \|V - Y\|^2/\xi \\
&\le 0.1\Delta_d^2 \cdot \mathrm{dist}(V, Y)^2/\xi \\
&\le \Delta_g^2/\xi
\end{aligned}$$

where the second step follows from $\xi = \mu k/n$, the third step follows from Definition of $\Delta_d$ (Definition D.5).

(In particular, the third step requires

$$\Delta_d \ge \Omega(\alpha^{-1.5}\mu^{1.5}k^2\gamma \cdot (\|W\|_{\infty,1}/n)^{1/2}) \quad (9)$$

)

**Random case.** We have $\eta = \mu k$. Using Lemma H.1, with high probability we know that $|T| = 0 \le \Delta_g^2/\xi$.

In the next analysis, we unify the SVD and random proofs into same way.

For each $i \in [n] \backslash T$, we have

$$\begin{aligned}
\|\widetilde{X}_{i,:}^\top - U_{i,:}^\top \Sigma V^\top Y\|_2^2 &= \|(U_{i,:}^\top \Sigma V^\top Y_\perp Y_\perp^\top D_{W_i} Y + N_{i,:}^\top D_i Y)(Y^\top D_{W_i} Y)^{-1}\|_2^2 \\
&\le \|(U_{i,:}^\top \Sigma V^\top Y_\perp Y_\perp^\top D_{W_i} Y + N_{i,:}^\top D_{W_i} Y)\|_2^2 \cdot \|(Y^\top D_{W_i} Y)^{-1}\|^2 \\
&\le 20\alpha^{-2}\eta^2 \cdot \|(U_{i,:}^\top \Sigma V^\top Y_\perp Y_\perp^\top D_{W_i} Y + N_{i,:}^\top D_{W_i} Y)\|_2^2 \\
&\le 20\alpha^{-2}\eta^2 \cdot 2(\|U_{i,:}^\top \Sigma V^\top Y_\perp Y_\perp^\top D_{W_i} Y\|_2^2 + \|N_{i,:}^\top D_{W_i} Y\|_2^2) \\
&\le 40\alpha^{-2}\eta^2 \cdot (\|U_{i,:}^\top\|_2^2\|\Sigma\|^2\|V^\top Y_\perp Y_\perp^\top D_{W_i} Y\|^2 + \|N_{i,:}^\top D_{W_i} Y\|_2^2) \\
&\le 40\alpha^{-2}\eta^2 \cdot (\frac{\mu k}{n}\|\Sigma\|^2\|V^\top Y_\perp Y_\perp^\top D_{W_i} Y\|^2 + \|N_{i,:}^\top D_{W_i} Y\|_2^2) \quad (10)
\end{aligned}$$

where the first step follows from Eq. (7), the second step follows from $\|Ax\|_2 \le \|A\| \cdot \|x\|_2$, the third step follows from $\sigma_{\min}(Y^\top D_{W_i} Y) \ge 0.25\alpha/\eta$ (for all $i \in [n] \backslash T$, see Eq. (8)), the fourth step follows from $(a+b)^2 \le 2a^2 + 2b^2$, the fifth step follows from $\|Ax\|_2 \le \|A\| \cdot \|x\|_2$, the sixth step follows from $\|U_{i,:}^\top\|^2 \le \mu k/n$.

Taking the summation over $i \in [n] \backslash T$ coordinates (for Eq. (10)), we have

$$\sum_{i \in [n] \backslash T} \|\widetilde{X}_{i,:}^\top - U_{i,:}^\top \Sigma V^\top Y\|_2^2 \le \sum_{i \in [n] \backslash T} 40\alpha^{-2}\eta^2 \cdot (\frac{\mu k}{n}\|\Sigma\|^2\|V^\top Y_\perp Y_\perp^\top D_{W_i} Y\|^2 + \|N_{i,:}^\top D_{W_i} Y\|_2^2) \quad (11)$$

For the first term in Eq. (11) (ignore coefficients $40\alpha^{-2}\eta^2$ and $\frac{\mu k}{n}\|\Sigma\|^2$), we have

$$\sum_{i \in [n] \backslash T} \|V^\top Y_\perp Y_\perp^\top D_{W_i} Y\|^2 \le n\gamma^2 \rho(Y) k^3 \, \mathrm{dist}(Y, V)^2$$

$$\leq n\gamma^2 \mu \sigma_{\min}^{-1}(\Sigma) k^3 \operatorname{dist}(Y, V)^2 \tag{12}$$

where the first step follows from Lemma F.2, the last step follows from $\rho(Y) \leq \mu/\sigma_{\min}(\Sigma)$.

For the second term in Eq. (11) (ignore coefficients $40\alpha^{-2}\eta^2$), we have

$$
\begin{aligned}
\sum_{i \in [n] \setminus T} \|N_{i,:}^\top D_{W_i} Y\|_2^2 &\leq \sum_{i \in [n]} \|N_{i,:}^\top D_{W_i} Y\|_2^2 \\
&= \|(W \circ N) Y\|_F^2 \\
&\leq k \|(W \circ N) Y\|^2.
\end{aligned}
\tag{13}
$$

where the last step follows from Fact A.7.

Loading Eq. (12) and Eq. (13) into Eq. (11), we have

$$
\sum_{i \in [n] \setminus T} \|\widetilde{X}_{i,:}^\top - U_{i,:}^\top \Sigma V^\top Y\|_2^2 \leq 40\alpha^{-2}\eta^2 \cdot \left(\frac{\mu k}{n} \|\Sigma\|^2 \cdot n\gamma^2 \mu \sigma_{\min}^{-1}(\Sigma) k^3 \operatorname{dist}(Y,V)^2 + k \|W \circ N\|^2\right)
$$

$$
\leq 40\alpha^{-2}\eta^2 \cdot (\mu^2 k^4 \tau \gamma^2) \cdot \operatorname{dist}(Y,V)^2 + 40\alpha^{-2}\eta^2 k \|W \circ N\|^2 \tag{14}
$$

where the last step follows from $\|\Sigma\| = 1$ and $\tau = \sigma_{\max}(\Sigma)/\sigma_{\min}(\Sigma)$.

Thus, we have

$$
\sum_{i \in [n] \setminus T} \|\widetilde{X}_{i,:}^\top - U_{i,:}^\top \Sigma V^\top Y\|_2^2 \leq 40\alpha^{-2}\eta^2 \cdot (\mu^2 k^4 \tau \gamma^2) \cdot \operatorname{dist}(Y,V)^2 + 40\alpha^{-2}\eta^2 k \|W \circ N\|^2
$$

$$
\leq 0.01\Delta_d^2 \cdot \operatorname{dist}(Y,V)^2 + 0.01\Delta_f^2 \cdot \|W \circ N\|^2
$$

$$
\leq 0.1\Delta_g^2
$$

where the first step follows from Eq. (14), the second step follows from Definition D.5, and the last step follows from Definition D.6.

(In particular, the second step above requires

$$\Delta_d \geq \Omega(\alpha^{-1}\eta\mu k^2 \tau^{0.5}\gamma) \tag{15}$$

and

$$\Delta_f \geq \Omega(\alpha^{-1}\eta k) \tag{16}$$

)

By Definition D.6 and choosing $\epsilon_{\mathrm{sk}}$ to be sufficiently small as Claim D.7, we know that

$$\Delta_g^2 \leq 0.01\Delta_u^2$$

Then, we can show

$$
\sum_{i \in [n] \setminus T} \|\vec{X}_{i,:}^\top - U_{i,:}^\top \Sigma V^\top Y\|_2^2 \leq 2 \sum_{i \in [n] \setminus T} \|\widetilde{X}_{i,:}^\top - U_{i,:}^\top \Sigma V^\top Y\|_2^2 + 2 \sum_{i \in [n] \setminus T} \|\vec{X}_{i,:}^\top - \widetilde{X}_{i,:}^\top\|_2^2
$$

$$
\leq \Delta_g^2,
$$

Note that

$$\|U_{i,:}^\top \Sigma V^\top Y\|_2^2 \leq \mu k/n = \xi.$$

If $\|\vec{X}_{i,:}^\top\|_2^2 \geq 4\xi$, then

$$\|\vec{X}_{i,:}^\top - U_{i,:}^\top \Sigma V^\top Y\|_2 \geq 2\sqrt{\xi} - \sqrt{\xi} \geq \sqrt{\xi}$$

which implies that

$$\|\vec{X}_{i,:}^\top - U_{i,:}^\top \Sigma V^\top Y\|_2^2 \geq \xi. \tag{17}$$

We define set $S \subset [n]$ as follows

$$S := \{i \in [n]\backslash T \mid \|\vec{X}_{i,:}^\top\|_2^2 \geq 4\xi\}.$$

Then we have

$$
\begin{aligned}
|S| &= |\{i \in [n]\backslash T \mid \|\vec{X}_{i,:}^\top\|_2^2 \geq 4\xi\}| \\
&\leq |\{i \in [n]\backslash T \mid \|\vec{X}_{i,:}^\top - U_{i,:}^\top \Sigma V^\top Y\|_2^2 \geq \xi\}| \\
&\leq \Delta_g^2/\xi.
\end{aligned}
$$

where the first step follows from the definition of $S$, the second step follows from Eq. (17), the third step follows from $\sum_{i \in [n]\backslash T} \|\vec{X}_{i,:}^\top - U_{i,:}^\top \Sigma V^\top Y\|^2 \leq \Delta_g^2$.

We can show

$$
\begin{aligned}
\|\widehat{X} - U\Sigma V^\top Y\|_F^2 &= \sum_{i=1}^n \|\widehat{X}_{i,:}^\top - U_{i,:}^\top \Sigma V^\top Y\|_2^2 \\
&= \sum_{i \in T \cup S} \|\widehat{X}_{i,:}^\top - U_{i,:}^\top \Sigma V^\top Y\|_2^2 + \sum_{i \notin T \cup S} \|\widehat{X}_{i,:}^\top - U_{i,:}^\top \Sigma V^\top Y\|_2^2 \\
&= \sum_{i \in T \cup S} \|\widehat{X}_{i,:}^\top - U_{i,:}^\top \Sigma V^\top Y\|_2^2 + \sum_{i \notin S} \|\vec{X}_{i,:}^\top - U_{i,:}^\top \Sigma V^\top Y\|_2^2 \\
&\leq \sum_{i \in T \cup S} 2(\|\widehat{X}_{i,:}^\top\|_2^2 + \|U_{i,:}^\top \Sigma V^\top Y\|_2^2) + \sum_{i \notin T \cup S} \|\vec{X}_{i,:}^\top - U_{i,:}^\top \Sigma V^\top Y\|_2^2 \\
&\leq |T \cup S| \cdot 2 \cdot (4\xi + \xi) + \Delta_g^2 \\
&= |T \cup S| \cdot 10\xi + \Delta_g^2 \\
&\leq 50\Delta g^2 \\
&\leq \Delta_u^2
\end{aligned}
$$

where the first step follows from the definition of $\|\widehat{X} - U\Sigma V^\top Y\|_F^2$, the second step follows from $S \subseteq [n]$, the third step follows from $\widehat{X}_{i,:}^\top = \vec{X}_{i,:}^\top$ when $i \notin S$, the fourth step follows from the triangle inequality, the fifth step follows from $\|\widehat{X}_{i,:}^\top\|^2 \leq 4\xi$ and $\|U_{i,:}^\top \Sigma V^\top Y\|^2 \leq \xi$, and the sixth step follows from $|T \cup S| \leq 2\Delta_g^2/\xi$, the last step follows from Claim D.7.

**Proof of Part 2.** We let $B = \Sigma V^\top Y$ and have

$$
\begin{aligned}
\sin\theta(U, X) &= \|U_\perp^\top X\| \\
&= \|U_\perp^\top (\overline{X} - UB)R^{-1}\| \\
&\leq \|(\overline{X} - UB)\| \cdot \|R^{-1}\| \\
&= \frac{\|(\overline{X} - UB)\|}{\sigma_{\min}(\overline{X})} \\
&\leq \frac{\Delta_u}{\sigma_{\min}(\overline{X})},
\end{aligned}
\tag{18}
$$

where the first step follows from the definition of

$$\sin\theta(U, X)$$

(see Definition A.10), the second step follows from

$$X = (\overline{X} - UB)R^{-1},$$

the 3rd step is due to the Cauchy-Schwarz inequality, and the 4th step is because of

$$\|R^{-1}\| = \frac{1}{\sigma_{\min}(\overline{X})},$$

and the 5th step follows from

$$\|(\overline{X} - U\Sigma V^\top Y)\|^2 \le \Delta_u^2,$$

which was proved in part 1, and it infers $\|(\overline{X} - UB)\| \le \Delta_u$ for $B = \Sigma V^\top Y$.

Using $\|(\overline{X} - UB)\| \le \Delta_u$, we have

$$
\begin{aligned}
\sigma_{\min}(\overline{X}) &\ge \sigma_{\min}(UB) - \Delta_u \\
&= \sigma_{\min}(U\Sigma V^\top Y) - \Delta_u \\
&= \sigma_{\min}(\Sigma V^\top Y) - \Delta_u \\
&\ge \sigma_{\min}(M^*)\cos\theta(Y, V) - \Delta_u \\
&\ge \sigma_{\min}(M^*)/2 - \Delta_u \\
&\ge \sigma_{\min}(M^*)/4,
\end{aligned}
\tag{19}
$$

where the second step follows from how we defined $B$, the third step follows from $U$ has orthonormal columns, the third step follows from

$$\sigma_{\min}(\Sigma V^\top Y) \ge \sigma_{\min}(M^*)\cos\theta(Y, V),$$

where the forth step follows from $\cos \ge 1/2$, and $\Delta_u \le \sigma_{\min}(M^*)/10$.

Then, by combining Eq. (18) and Eq. (19), we have

$$
\begin{aligned}
\sin\theta(U, X) &\le \frac{1}{\sigma_{\min}(\overline{X})}\Delta_u \\
&\le 4\Delta_u/\sigma_{\min}(M^*).
\end{aligned}
\tag{20}
$$

Therefore,

$$
\begin{aligned}
\mathrm{dist}(U, X) &\le 2\sin\theta(U, X) \\
&\le 8\Delta_u/\sigma_{\min}(M^*).
\end{aligned}
$$

where the first step follows from Part 5 of Lemma A.14, and the last step follows from Eq. (20).

**Proof Part 3.** Given $\overline{X} = XR$, we have $\overline{X}_{i,:}^\top = X_{i,:}^\top R$ and

$$
\begin{aligned}
\|X_{i,:}\|_2^2 &\le \|\overline{X}_{i,:}\|_2^2\|R^{-1}\|^2 \\
&\le \frac{\xi}{\sigma_{\min}(\overline{X})^2} \\
&\le \frac{\xi}{(\sigma_{\min}(M^*) - 2\Delta_u)^2} \\
&\le 8\xi/\sigma_{\min}(M^*)^2,
\end{aligned}
\tag{21}
$$

where the first step follows from $\overline{X}_{i,:}^\top = X_{i,:}^\top R$ and Cauchy-Schwarz inequality, the second step follows from $\|\overline{X}_{i,:}\|^2 \le \xi$ and $\|R^{-1}\| = \frac{1}{\sigma_{\min}(\overline{X})}$, the third step follows from $\frac{1}{\sigma_{\min}(\overline{X})^2} \le \frac{1}{(\sigma_{\min}(M^*)-2\Delta_u)^2}$, and the last step follows from $\Delta_u \le 0.1\sigma_{\min}(M^*)$.

To see this, we have

$$
\begin{aligned}
\|R^{-1}\|^2 &= \lambda_{\max}(R^{-1}(R^{-1})^\top) \\
&= \lambda_{\max}((\overline{X}^\top\overline{X})^{-1}) \\
&= \frac{1}{\lambda_{\min}(\overline{X}^\top\overline{X})} \\
&= \frac{1}{\sigma_{\min}^2(\overline{X})},
\end{aligned}
$$

where the first step follows from $\|A\|^2 = \lambda_{\max}(AA^\top)$, the second step follows from $\overline{X}^\top \overline{X} = R^\top R$, the third step follows from $\lambda_{\max}(A^{-1}) = \lambda_{\min}(A)^{-1}$, and the last step follows from $\lambda(A^\top A) = \sigma^2(A)$.

Then,

$$
\begin{aligned}
\rho(X) &= \max_{i \in [n]} \frac{n}{k} \|X_{i,:}\|_2^2 \\
&= \frac{n}{k} \max_{i \in [n]} \|X_{i,:}\|_2^2 \\
&\leq \frac{n}{k} \frac{8\xi}{\sigma_{\min}(M^*)^2} \\
&= \frac{8\mu}{\sigma_{\min}(M^*)^2},
\end{aligned}
$$

where the first step follows from the definition of $\rho(X)$, the second step follows from simple algebra, the third step follows from Eq. (21), and the last step follows from $\frac{\xi n}{k} = \mu$. □

## E  THE ANALYSIS OF THE INDUCTION LEMMA

The goal of this section is to prove induction.

**Lemma E.1.** *Suppose*

$$
\Delta_u \leq \Delta_d \cdot \mathrm{dist}(Y_t, V) + \Delta_f \cdot \|W \circ N\|.
$$

*and*

$$
\Delta_d \leq \frac{1}{100}\sigma_{\min}(M^*)
$$

*For any $t \geq 1$,*

$$
\mathrm{dist}(X_t, U) \leq \frac{1}{2^t} + 100\sigma_{\min}(M^*)^{-1} \cdot \Delta_f \cdot \|W \circ N\|,
$$
$$
\mathrm{dist}(Y_t, V) \leq \frac{1}{2^t} + 100\sigma_{\min}(M^*)^{-1} \cdot \Delta_f \cdot \|W \circ N\|, \tag{22}
$$

*Proof.* By using induction, we show that Eq. (22) holds.

**Base case:** By Lemma F.3, $Y_1$ satisfies

$$
\mathrm{dist}(Y_1, V) \leq \frac{1}{2} + 8\sigma_{\min}(M^*)^{-1} \cdot \Delta_f \cdot \|W \circ N\|
$$

**Inductive case:** Suppose that it holds for the first $t$ cases. By definition of $\Delta_u$, we have

$$
\Delta_u \leq \Delta_d \cdot \mathrm{dist}(Y_t, V) + \Delta_f \cdot \|W \circ N\| \tag{23}
$$

We have

$$
\begin{aligned}
&\mathrm{dist}(X_{t+1}, U) \\
&\leq \frac{8}{\sigma_{\min}(M^*)} \cdot \Delta_u \\
&= \frac{8}{\sigma_{\min}(M^*)} \cdot (\Delta_d \cdot \mathrm{dist}(Y_t, V) + \Delta_f \cdot \|W \circ N\|) \\
&\leq \frac{1}{2}\mathrm{dist}(Y_t, V) + 8\sigma_{\min}(M^*)^{-1} \cdot \Delta_f \cdot \|W \circ N\|
\end{aligned}
$$

where the first step follows from Part 2 of Lemma D.9, the second step follows from Eq. (23) the last step follows from $\Delta_d/\sigma_{\min}(M^*) \leq 1/100$. □

# F  Tools From Previous Work

In this section, we state several tools from previous work. In Section F.1, we introduce a varied version of a Lemma from Li et al. (2016) to bound the eigenvalues. In Section F.2, we bound $\|V^\top Y_\perp (Y_\perp)^\top D_{W_i} Y\|^2$. In Section F.3, we summarize the base case lemma.

## F.1  Bounding Eigenvalues

Now, in this section, we start to bound the eigenvalues.

**Lemma F.1** (A variation of Lemma 10 in Li et al. (2016)). *Let $Y$ be a (column) orthogonal matrix in $\mathbb{R}^{n \times k}$. Let $\epsilon \in (0,1)$. We have*

$$|\{i \in [n] \mid \sigma_{\min}(Y^\top D_{W_i} Y) \leq (1-\epsilon)\alpha\}| \leq 10^4 \cdot \frac{\mu^2 k^3 \gamma^2}{\epsilon^4 \alpha^3} \cdot \|W\|_{\infty,1} \cdot \|V - Y\|^2.$$

*Proof.* Let $j$ be an arbitrary integer in $[n]$. Let $g$ be greater than 0. $j$ is called "good" if

$$\|Y_j - V_j\|_2^2 \leq g^2.$$

We define $S_g \subset [n]$ as follows

$$S_g := \{j \in [n] \mid \|Y_j - V_j\|_2^2 \leq g^2\}.$$

For convenience, we define $\overline{S}_g \subset [n]$ as follows

$$\overline{S}_g := [n] \backslash S_g.$$

We choose $g$ to satisfy the following condition

$$g^2 = \frac{\epsilon^2 \alpha}{20\|W\|_{\infty,1}}. \tag{24}$$

Let $a$ be an arbitrary unit vector in $\mathbb{R}^k$. Thus, we have

$$\begin{aligned}
a^\top Y^\top D_{W_i} Y a &= \sum_{j \in [n]} (D_{W_i})_j \langle a, Y_j \rangle^2 \\
&\geq \sum_{j \in S_g} (D_{W_i})_j \langle a, Y_j \rangle^2 \\
&= \sum_{j \in S_g} (D_{W_i})_j (\langle a, V_j \rangle + \langle a, Y_j - V_j \rangle)^2 \\
&\geq (1 - \epsilon/4) \sum_{j \in S_g} (D_{W_i})_j \langle a, V_j \rangle^2 - 4\epsilon^{-1} \sum_{j \in S_g} (D_{W_i})_j \langle a, Y_j - V_j \rangle^2 \\
&\geq (1 - \epsilon/4) \sum_{j \in S_g} (D_{W_i})_j \langle a, V_j \rangle^2 - 4\epsilon^{-1} g^2 \sum_{j \in S_g} (D_{W_i})_j \\
&\geq (1 - \epsilon/4) \sum_{j \in S_g} (D_{W_i})_j \langle a, V_j \rangle^2 - 4\epsilon^{-1} g^2 \sum_{j \in [n]} (D_{W_i})_j \\
&\geq (1 - \epsilon/4) \sum_{j \in [n]} (D_{W_i})_j \langle a, V_j \rangle^2 - \sum_{j \in \overline{S}_g} (D_{W_i})_j \langle a, V_j \rangle^2 - 4\epsilon^{-1} g^2 \sum_{j \in [n]} (D_{W_i})_j \\
&\geq (1 - \epsilon/4) \sum_{j \in [n]} (D_{W_i})_j \langle a, V_j \rangle^2 - \frac{\mu k}{n} \sum_{j \in \overline{S}_g} (D_{W_i})_j - 4\epsilon^{-1} g^2 \sum_{j \in [n]} (D_{W_i})_j, \tag{25}
\end{aligned}$$

where the first step follows from simple algebra, the second step follows from $S_g \subset [n]$, the third step follows from the property of the inner product, the fourth step follows from Fact A.5, the fifth step follows from the definition of $S_g$, the sixth step follows from $(D_{W_i})_j \geq 0$, the

seventh step follows from $1 - \epsilon/4 \leq 1$, and the last step follows from the property of $V$ (e.g. $\langle a, V_j \rangle^2 \leq \|a\|_2^2 \cdot \|V_j\|_2^2 \leq \|V_j\|_2^2 \leq \xi \leq \mu k/n$).

We can show that

$$
\begin{aligned}
\sum_{j \in [n]} (D_{W_i})_j \langle a, V_j \rangle^2 &= a^\top V^\top (D_{W_i})_j V a \\
&\geq \sigma_{\min}(V^\top (D_{W_i})_j V) \\
&\geq \alpha,
\end{aligned}
\tag{26}
$$

where the second step follows from Fact A.7 and the third step follows from definition of $\alpha$ (see Definition 3) and $\sigma_{\min}(A) \leq \sigma_{\min}(B)$ if $A \preceq B$.

Moreover, recall

$$
\|W\|_{\infty,1} = \max_{i \in [n]} \sum_{j \in [n]} |(D_{W_i})_j|,
\tag{27}
$$

We can show that

$$
\begin{aligned}
4\epsilon^{-1} g^2 \sum_{j \in [n]} (D_{W_i})_j &\leq 4\epsilon^{-1} g^2 \|W\|_{\infty,1} \\
&= 4\epsilon^{-1} \frac{\epsilon^2 \alpha}{20\|W\|_{\infty,1}} \|W\|_{\infty,1} \\
&\leq \frac{\epsilon \alpha}{4},
\end{aligned}
\tag{28}
$$

where the first step follows from the definition of $\|W\|_{\infty,1}$ (see Eq. (27)), the second step follows from the definition of $g^2$ (see Eq. (24)), and the last step follows from simple algebra.

We define

$$
T := \{i \in [n] \mid \sigma_{\min}(Y^\top D_{W_i} Y) \leq (1-\epsilon)\alpha\}.
$$

Let us consider

$$
\sum_{j \in \overline{S}_g} (D_{W_i})_j.
$$

We define

$$
S := \{i \in [n] \mid \frac{\mu k}{n} \sum_{j \in \overline{S}_g} (D_{W_i})_j \geq \frac{\epsilon \alpha}{4}\}.
\tag{29}
$$

If $i \notin S$, then we have

$$
\begin{aligned}
a^\top Y^\top D_{W_i} Y a &\geq (1-\epsilon/4) \sum_{j \in [n]} (D_{W_i})_j \langle a, V_j \rangle^2 - \frac{\mu k}{n} \sum_{j \in \overline{S}_g} (D_{W_i})_j - 4\epsilon^{-1} g^2 \sum_{j \in [n]} (D_{W_i})_j \\
&\geq (1-\epsilon/4)\alpha - \frac{\mu k}{n} \sum_{j \in \overline{S}_g} (D_{W_i})_j - 4\epsilon^{-1} g^2 \sum_{j \in [n]} (D_{W_i})_j \\
&\geq (1-\epsilon/4)\alpha - \epsilon\alpha/4 - 4\epsilon^{-1} g^2 \sum_{j \in [n]} (D_{W_i})_j \\
&\geq (1-\epsilon/4)\alpha - \epsilon\alpha/4 - \epsilon\alpha/4 \\
&\geq (1-\epsilon)\alpha,
\end{aligned}
$$

where the first step follows from Eq. (25), the second step follows Eq. (26), the third step follows from the Definition of $S$ (see Eq. (29)), the fourth step follows from Eq. (28), and the last step follows from simple algebra.

In summary, we know that if $i \notin S$, then $i \notin T$. By taking its contraposition, we have that if $i \in T$, then $i \in S$.

Thus, we can show that

$$|T| \leq |S|$$

In the next a few paragraphs, we will explain how to upper bound $|S|$.

Using Fact A.7

$$\sum_{j \in [n]} \|V_j - Y_j\|_2^2 = \|V - Y\|_F^2,$$

By simple counting argument (you have $n$ positive values, their summation is $\|V - Y\|_F^2$, you can't have more than $\|V - Y\|_F^2/g^2$ of them that are bigger than $g^2$), we have

$$|\overline{S}_g| \leq \|V - Y\|_F^2/g^2. \tag{30}$$

Let $u_S \in \mathbb{R}^n$ be the indicator vector of $S$, i.e.,

$$\forall i \in [n], \quad (u_S)_i = \begin{cases} 1 & \text{if } i \in S; \\ 0 & \text{otherwise } i \notin S. \end{cases}$$

Let $u_g \in \mathbb{R}^n$ be the indicator vector of $\overline{S}_g$, i.e.,

$$\forall i \in [n], \quad (u_g)_i = \begin{cases} 1 & \text{if } i \in \overline{S}_g; \\ 0 & \text{otherwise } i \notin \overline{S}_g. \end{cases}$$

Then, we know that

$$\begin{aligned} u_S^\top W u_g &= \sum_{i \in S} \sum_{j \in \overline{S}_g} (D_{W_i})_j \\ &\geq |S| \cdot \min_{i \in S} \sum_{j \in \overline{S}_g} (D_{W_i})_j \\ &\geq |S| \cdot \frac{\epsilon \alpha n}{4\mu k}, \end{aligned} \tag{31}$$

where the first step follows from simple algebra, and the second step follows from simple algebra, the third step follows from Definition (29).

On the other hand,

$$\begin{aligned} u_S^\top W u_g &= u_S^\top \mathbf{1}_n \mathbf{1}_n^\top u_g + u_S^\top (W - \mathbf{1}_n \mathbf{1}_n^\top) u_g \\ &= \|u_S\|_1 \cdot \|u_g\|_1 + u_S^\top (W - \mathbf{1}_n \mathbf{1}_n^\top) u_g \\ &\leq \|u_S\|_1 \cdot \|u_g\|_1 + \|u_S\|_2 \cdot \|W - \mathbf{1}_n \mathbf{1}_n^\top\| \cdot \|u_g\|_2 \\ &\leq \|u_S\|_1 \cdot \|u_g\|_1 + \gamma n \cdot \|u_S\|_2 \cdot \|u_g\|_2 \\ &\leq |S||\overline{S}_g| + \gamma n \sqrt{|S||\overline{S}_g|}, \end{aligned} \tag{32}$$

where the first step follows from simple algebra, the second step follows from simple algebra, the third step follows from $x^\top A y \leq \|x\|_2 \|A\| \|y\|_2$, the fourth step follows from Definition 2, and the last step follows from $u_S$ and $u_g$ are indicator vectors.

By combining Eq. (31) and Eq. (32), we have

$$|\overline{S}_g| + \gamma n \cdot \sqrt{|\overline{S}_g|/|S|} \geq \frac{\epsilon \alpha n}{4\mu k}. \tag{33}$$

Note that if $A + B \geq C$. Then if $A \leq C/2$, then $B \geq C/2$.

For the terms in Eq. (33), we define $A$, $B$ and $C$ as follows

$$A := |\overline{S}_g|$$

$$B := \gamma n \cdot \sqrt{|\overline{S}_g|/|S|}$$

$$C := \frac{\epsilon \alpha n}{4\mu k}$$

If

$$|\overline{S}_g| \leq \frac{\epsilon \alpha n}{8\mu k},$$

then, we have

$$\gamma n \cdot \sqrt{|\overline{S}_g|/|S|} \geq \frac{\epsilon \alpha n}{8\mu k}.$$

The above equation implies

$$
\begin{aligned}
|S| &\leq 500 \cdot \frac{\mu^2 k^2 \gamma^2}{\epsilon^2 \alpha^2} \cdot |\overline{S}_g| \\
&\leq 500 \cdot \frac{\mu^2 k^2 \gamma^2}{\epsilon^2 \alpha^2} \cdot (\|V - Y\|_F^2 / g^2), \\
&\leq 500 \cdot \frac{\mu^2 k^2 \gamma^2}{\epsilon^2 \alpha^2} \cdot \|V - Y\|_F^2 \cdot \frac{20\|W\|_{\infty,1}}{\epsilon^2 \alpha}
\end{aligned}
\tag{34}
$$

where the second step follows from Eq. (30) and last step follows from Eq. (24).

We have

$$
\begin{aligned}
|\{i \in [n] \mid \sigma_{\min}(Y^\top D_{W_i} Y) \leq (1-\epsilon)\alpha\}| &\leq |S| \\
&\leq 10^4 \cdot \frac{\mu^2 k^2 \gamma^2}{\epsilon^4 \alpha^3} \cdot \|W\|_{\infty,1} \cdot \|V - Y\|_F^2 \\
&\leq 10^4 \cdot \frac{\mu^2 k^2 \gamma^2}{\epsilon^4 \alpha^3} \cdot \|W\|_{\infty,1} \cdot k \cdot \|V - Y\|^2
\end{aligned}
$$

where the first step follows from the definition of $S$ and the second step follows from combining Eq. (34) and Eq. (24), and the last step follows from Fact A.7. This completes the proof. $\square$

## F.2 BOUNDING $\|V^\top Y_\perp (Y_\perp)^\top D_{W_i} Y\|^2$

In this section, we bound $\|V^\top Y_\perp (Y_\perp)^\top D_{W_i} Y\|^2$ by a multiplicative factor times $\|Y - V\|^2$.

**Lemma F.2** (A variation of Lemma 11 in Li et al. (2016)). *Let $Y$ be a (column) orthogonal matrix in $\mathbb{R}^{n \times k}$. Let $i \in [n]$. Then we have*

$$\sum_{i \in [n]} \|V^\top Y_\perp (Y_\perp)^\top D_{W_i} Y\|^2 \leq \gamma^2 \rho(Y) n k^3 \|Y - V\|^2$$

For the completeness, we still provide the proof.

*Proof.* Let $j', j$ be two positive integers in $[k]$. $Y_j$ represents the matrix $Y$'s $j$-th column. $\widetilde{V}_j$ represents the matrix $Y_\perp Y_\perp^\top V$'s $j$-th column. We define $x^{j,j'} \in \mathbb{R}^n$ as

$$x_i^{j,j'} = (\widetilde{V}_j)_i (Y_{j'})_i.$$

We need to show that the spectral norm of

$$V^\top Y_\perp Y_\perp^\top D_{W_i} Y,$$

is bounded. Note that $\langle \widetilde{V}_j, Y_{j'} \rangle = 0$, which implies that

$$\sum_{i \in [n]} x_i^{j,j'} = 0.$$

For $V_j^\top Y_\perp Y_\perp^\top D_{W_i} Y_{j'}$,

$$V_j^\top Y_\perp Y_\perp^\top D_{W_i} Y_{j'} = \sum_{s \in [n]} (D_{W_i})_s (\widetilde{V}_j)_s (Y_{j'})_s$$

$$= \sum_{s \in [n]} (D_{W_i})_s x_s^{j,j'}, \tag{35}$$

where the first step follows from the definition of

$$V_j^\top Y_\perp Y_\perp^\top D_{W_i} Y_{j'}$$

and the second step follows from $(\widetilde{V}_j)_s (Y_{j'})_s = x_s^{j,j'}$.

It implies that

$$\sum_{i \in [n]} (\sum_{s \in [n]} (D_{W_i})_s x_s^{j,j'})^2 = \|W x^{j,j'}\|_2^2$$

$$= \|(W - \mathbf{1}_n \mathbf{1}_n^\top) x^{j,j'}\|_2^2$$

$$\leq \|W - \mathbf{1}_n \mathbf{1}_n^\top\|^2 \|x^{j,j'}\|_2^2$$

$$\leq \gamma^2 n^2 \|x^{j,j'}\|_2^2,$$

where the first step follows from the definition of $\|W x^{j,j'}\|_2^2$, the second step follows from $\mathbf{1}_n \mathbf{1}_n^\top x^{j,j'} = 0$, the third step follows from $\|Ax\|_2 \leq \|A\| \|x\|_2$, and the last step follows from $\|W - \mathbf{1}_n \mathbf{1}_n^\top\| \leq \gamma n$ (see Definition 2).

Observe that

$$\|x^{j,j'}\|_2^2 = \sum_{i \in [n]} (x^{j,j'})^2$$

$$= \sum_{i \in [n]} (\widetilde{V}_j)_i^2 (Y_{j'})_i^2$$

$$\leq \frac{\rho(Y)k}{n} \sum_{i \in [n]} (\widetilde{V}_j)_i^2$$

$$= \frac{\rho(Y)k}{n} \|\widetilde{V}_j\|_2^2$$

$$\leq \frac{\rho(Y)k}{n} \|Y_\perp Y_\perp^\top V\|^2$$

$$= \frac{\rho(Y)k}{n} \|Y_\perp Y_\perp^\top (Y - V)\|^2$$

$$= \frac{\rho(Y)k}{n} \|Y_\perp Y_\perp^\top\| \cdot \|Y - V\|^2$$

$$\leq \frac{\rho(Y)k}{n} \|Y - V\|^2$$

where the first step follows from the definition of $\|x^{j,j'}\|_2^2$, the second step follows from $(x^{j,j'})^2 = (\widetilde{V}_j)_i^2 (Y_{j'})_i^2$, the third step follows from definition of $\rho$, the fourth step follows from the definition of $\|\cdot\|_2^2$, the fifth step follows from the fact that $\widetilde{V}_j$ is defined to be the $j$-th column of $Y_\perp Y_\perp^\top V$, the sixth step follows from $Y_\perp^\top Y = 0$, the seventh step follows from $\|AB\| \leq \|A\| \cdot \|B\|$, and the last step follows from $\|Y_\perp Y_\perp^\top\| \leq 1$.

It implies that

$$\sum_{i\in[n]}(\sum_{s\in[n]}(D_{W_i})_s x_s^{j,j'})^2 \le \gamma^2 \rho(Y) n k \|Y - V\|^2. \tag{36}$$

Now we are ready to bound $V^\top Y_\perp Y_\perp^\top D_{W_i} Y$. Note that

$$\begin{aligned}
\|V^\top Y_\perp Y_\perp^\top D_{W_i} Y\|^2 &\le \|V^\top Y_\perp Y_\perp^\top D_{W_i} Y\|_F^2 \\
&\le \sum_{j,j'\in[k]} (V_j^\top Y_\perp Y_\perp^\top D_{W_i} Y_{j'})^2 \\
&= \sum_{j,j'\in[k]} (\sum_{s\in[n]}(D_{W_i})_s x_s^{j,j'})^2, 
\end{aligned} \tag{37}$$

where the first step follows from $\|A\| \le \|A\|_F$ for all matrix $A$, the second step follows from definition of $\|\cdot\|_F$, and the third step follows from Eq. (35).

This implies that

$$\begin{aligned}
\sum_{i\in[n]} \|V^\top Y_\perp Y_\perp^\top D_{W_i} Y\|^2 &\le \sum_{i\in[n]}\sum_{j,j'\in[k]}(\sum_{s\in[n]}(D_{W_i})_s x_s^{j,j'})^2 \\
&\le \sum_{j,j'\in[k]} \gamma^2 \rho(Y) n k \|Y - V\|^2 \\
&= \sum_{j\in[k]}\sum_{j'\in[k]} \gamma^2 \rho(Y) n k \|Y - V\|^2 \\
&= kk\gamma^2 \rho(Y) n k \|Y - V\|^2 \\
&= \gamma^2 \rho(Y) n k^3 \|Y - V\|^2,
\end{aligned}$$

where the first step follows from Eq. (37), the second step follows from Eq. (36), the third step follows from simple algebra, the fourth step follows from the property of $\sum$ (e.g. $\sum_{i=1}^n a = an$), and the last step follows from simple algebra. This completes the proof. $\qquad\square$

### F.3 SUMMARY OF BASE CASE LEMMA

We state a general base case lemma that covers both random initialization and SVD initialization.

**Lemma F.3** (General base case lemma). *Let $\Delta_f := 20\alpha^{-1}\eta k$. For the base case, we have*

$$\text{dist}(Y_1, V) \le \frac{1}{2} + 8\sigma_{\min}(M^*)^{-1} \cdot \Delta_f \cdot \|W \circ N\|_F.$$

The proofs are delayed into Section H and Section G in which we analyze the random and SVD initializations with different parameters.

## G SVD INITIALIZATION AND MAIN RESULT

In Section G.1, we introduce our assumption on $\delta$. In Section G.2, we bound $\|(W - \mathbf{1}_n \mathbf{1}_n^\top) \circ H\|$. In Section G.3, we analyze the property of the rank-$k$ SVD. In Section G.4, we analyze the properties of dist and $\rho$. In Section G.5, we present our main result.

### G.1 ASSUMPTION

Here, we set the parameter $\delta$.

**Definition G.1.** We assume that

$$\delta := 0.001 \cdot \|W \circ N\| \le \alpha\sigma_{\min}(M^*)/k.$$

## G.2 Bounding $\|(W - \mathbf{1}_n\mathbf{1}_n^\top) \circ H\|$

In this section, we bound $\|(W - \mathbf{1}_n\mathbf{1}_n^\top) \circ H\|$ in terms $\gamma$, the rank $k$ and the top singular value $\sigma_1$.

**Lemma G.2** (Spectral lemma, Lemma 5 in Li et al. (2016)). *Let $K$ and $J$ be (column) orthogonal matrices, whose sizes are $n \times n$. Let $H$ be an arbitrary matrix in $\mathbb{R}^{n \times n}$ satisfying*

$$H = A\Sigma B^\top, \tag{38}$$

*where $A \in \mathbb{R}^{n \times k}$, $B \in \mathbb{R}^{n \times k}$, and $\Sigma \in \mathbb{R}^{k \times k}$. Note that $A$ and $B$ might not be orthogonal, but $\Sigma$ is diagonal. The matrix $W \in \mathbb{R}^{n \times n}$ is an entry wise non-negative matrix, which has an artificial spectral gap, satisfying*

$$W = \mathbf{1}_n\mathbf{1}_n^\top + \gamma n J \Sigma_W K^\top,$$

*where*

$$\|\Sigma_W\| = 1.$$

*Let $\sigma_1 := \max_{r \in [k]} \sigma_r(\Sigma)$. Then, we have*

$$\|(W - \mathbf{1}_n\mathbf{1}_n^\top) \circ H\| \le \gamma k \sigma_1 \sqrt{\rho(A)\rho(B)}.$$

*Proof.* For each $i \in [n]$, let $A_{i,:}$ denote the $i$-th row of matrix $A \in \mathbb{R}^{n \times k}$. For each $r \in [k]$, let $A_r$ denote the $r$-th column of matrix $A \in \mathbb{R}^{n \times k}$. Then we have

$$
\begin{aligned}
\sum_{r=1}^{k} \|A_r \circ x\|_2^2 &= \sum_{r=1}^{k}\sum_{i=1}^{n} A_{i,r}^2 x_i^2 \\
&= \sum_{i=1}^{n} x_i^2 \sum_{r=1}^{k} A_{i,r}^2 \\
&= \sum_{i=1}^{n} x_i^2 \|A_{i,:}\|_2^2 \\
&\le \sum_{i=1}^{n} x_i^2 \frac{k}{n}\rho(A) \\
&\le \frac{k}{n}\rho(A) \tag{39}
\end{aligned}
$$

where the first step follows from the definition of $\|\cdot\|_2^2$, the second step follows from simple algebra, the third step follows from the definition of $\|\cdot\|_2^2$, the fourth step follows from the definition of $\rho$ (Definition A.8), and the last step follows from $\sum_{i=1}^{n} x_i^2 \le 1$.

Let $x, y \in \mathbb{R}^n$ be two arbitrary unit vectors. Then, we have

$$
\begin{aligned}
x^\top((W - \mathbf{1}_n\mathbf{1}_n^\top) \circ H)y &= x^\top((W - \mathbf{1}_n\mathbf{1}_n^\top) \circ (A\Sigma B^\top))y \\
&= \sum_{r=1}^{k} \sigma_r x^\top((W - \mathbf{1}_n\mathbf{1}_n^\top) \circ A_r B_r^\top)y \\
&= \gamma n \sum_{r=1}^{k} \sigma_r (A_r \circ x)^\top J\Sigma_W K^\top (B_r \circ y) \\
&\le \gamma n \sum_{r=1}^{k} \sigma_r \|A_r \circ x\|_2 \cdot \|J\Sigma_W K^\top\| \cdot \|B_r \circ y\|_2 \\
&\le \gamma n \sum_{r=1}^{k} \sigma_r \|A_r \circ x\|_2 \cdot \|B_r \circ y\|_2 \\
&\le \gamma n \sigma_1 \sum_{r=1}^{k} \|A_r \circ x\|_2 \cdot \|B_r \circ y\|_2
\end{aligned}
$$

$$\leq \gamma n \sigma_1 (\sum_{r=1}^{k} \|A_r \circ x\|_2^2)^{1/2} (\sum_{r=1}^{k} \|B_r \circ y\|_2^2)^{1/2}$$

$$\leq \gamma n \sigma_1 \sqrt{\frac{k}{n}\rho(A)} \sqrt{\frac{k}{n}\rho(B)}$$

$$\leq \gamma \sigma_1 \cdot k \cdot (\rho(A)\rho(B))^{1/2},$$

where the first step follows from the definition of $H$ (see Eq. (38)), the second step follows from $\Sigma$ is a diagonal matrix, the third step follows from the definition of $W$, the fourth step follows from $\|Ax\|_2 \leq \|A\| \cdot \|x\|_2$, the fifth step follows from $\|J\Sigma_W K^\top\| \leq \|J\| \cdot \|\Sigma_W\| \cdot \|W\| \leq 1$, the sixth step follows from $\sigma_1 = \max_{r \in [k]} \sigma_r$, the seventh step follows from Cauchy-Schwarz inequality, the eighth step follows from Eq. (39), and the last step follows from simple algebra.

The lemma follows from the definition of the operator norm. $\qquad\square$

**Lemma G.3** (Wedin's Theorem, Lemma 6 in Li et al. (2016)). *$M^*$ is a matrix, and $\sigma_1, \ldots, \sigma_n$ are the singular values of $M^*$. $\widetilde{M}$ is a matrix, and $\widetilde{\sigma}_1, \ldots, \widetilde{\sigma}_n$ are the singular values of $\widetilde{M}$. Suppose that $X, Y$ and $U, V$ are the first $k$ singular vectors (left and right) of $\widetilde{M}, M^*$ respectively. If there exists $a$ which is greater than $0$ and satisfies*

$$\max_{r \in \{k+1, \cdots, n\}} \widetilde{\sigma}_r \leq \min_{i \in \{1, \cdots, k\}} \sigma_i - a,$$

*then*

$$\frac{\|M^* - \widetilde{M}\|}{a} \geq \max\{\sin\theta(V, Y), \sin\theta(U, X)\}.$$

### G.3 PROPERTY OF RANK-$k$ SVD

Now, we first define the parameter $\Delta_1$, and then analyze the properties of rank-$k$ SVD.

**Definition G.4.** We define $\Delta_1$ as follows

$$\Delta_1 := \frac{10(\gamma\mu k + \delta)}{\sigma_{\min}(M^*)}.$$

**Lemma G.5** (Lemma 7 in Li et al. (2016)). *Assume that $W$ and $M^*$ satisfy every assumption. We define $(X, \Sigma, Y) := \text{rank-}k \text{ SVD}(W \circ M)$. Let $\Delta_1$ be defined as Definition G.4 and assume that $\Delta_1 \leq 0.01$. Then, we have*

$$\max\{\tan\theta(X, U), \tan\theta(Y, V)\} \leq 0.5\Delta_1.$$

*Proof.* We know that

$$\begin{aligned}
\|W \circ M - M^*\| &= \|W \circ (M^* + N) - M^*\| \\
&\leq \|W \circ M^* - M^*\| + \|W \circ N\| \\
&= \|W \circ M^* - M^*\| + \delta \\
&\leq \gamma\mu k \sigma_{\max}(M^*) + \delta \\
&\leq \gamma\mu k + \delta
\end{aligned} \tag{40}$$

where the first step follows from the definition of $M$, the second step follows from triangle inequality and the third step follows from Definition G.1, the fourth step follows from Lemma G.2, the last step follows from $\sigma_{\max}(M^*) = 1$.

Therefore,

$$\begin{aligned}
\max_{r \in [k+1, n]} \sigma_r(W \circ M) &\leq \max_{r \in [k+1, n]} \sigma_r(W \circ M - M^*) + \max_{r \in [k+1, n]} \sigma_r(M^*) \\
&\leq \max_{r \in [k+1, n]} \sigma_r(W \circ M - M^*) + 0 \\
&\leq \|W \circ M - M^*\| \\
&\leq \gamma\mu k + \delta
\end{aligned}$$

$$\leq \frac{1}{4}\sigma_{\min}(M^*) + \delta$$

$$\leq \frac{1}{2}\sigma_{\min}(M^*),$$

where the first step follows from triangle inequality, the second step follows from the fact that $M^*$ has rank-$k$, the third step follows from Eq. (40), the fourth step follows from $\gamma\mu k < 0.1\sigma_{\min}(M^*)$, and the last step follows from $\delta \leq 0.1\sigma_{\min}(M^*)$.

Now, by Wedin's theorem (see Lemma G.3) with

$$a = \frac{1}{2}\sigma_{\min}(M^*),$$

for

$$(X, \Sigma, Y) = \text{rank-}k \, \text{SVD}(W \circ M),$$

we have

$$\max\{\sin\theta(U, X), \sin\theta(V, Y)\} \leq \frac{2(\gamma\mu k + \delta)}{\sigma_{\min}(M^*)}.$$

By our choice of parameters

$$\sin\theta \leq 1/2.$$

Using Lemma A.14, we have

$$\tan\theta \leq 2\sin\theta.$$

Then the lemma follows. $\qquad\square$

### G.4 Initial Properties of Distance and $\rho$

We analyze the properties of distance and $\rho$ during initialization. We show that as long as $\Delta_1$ is chosen properly, the distance and $\rho$ can be bounded.

**Lemma G.6** ((SVD initialization, a variation of Lemma 8 in Li et al. (2016)). *Assume that $W$ and $M^*$ satisfy every assumption. Let $\Delta_1$ be defined as Defintion G.4. Assume that $\Delta_1 \leq 0.01/k$. Then, we have*

- *Part 1.* $\text{dist}(V, Y_1) \leq 1/2$.

- *Part 2. Let $\rho(\cdot)$ be defined as Definition A.8. We have $\rho(Y_1) \leq 4\mu$.*

*Proof.* **Proof of Part 1.** First, we consider $\widetilde{Y}_1 \in \mathbb{R}^{n \times k}$. By Lemma G.5, we get that

$$\text{dist}(\widetilde{Y}_1, V) \leq \Delta_1,$$

which means that there exists $Q \in O^{k \times k}$, such that

$$\|\widetilde{Y}_1 Q - V\| \leq \Delta_1. \tag{41}$$

Hence,

$$\|\widetilde{Y}_1 Q - V\|_F \leq \sqrt{k} \cdot \|\widetilde{Y}_1 Q - V\|$$
$$\leq \sqrt{k} \cdot \Delta_1$$
$$\leq \frac{1}{10}, \tag{42}$$

where the first step follows from Fact A.7, the second step follows from Eq. (41), and the last step follows from $\Delta_1 \leq 0.01/k$ .

Next, we consider $\overline{Y_1} \in \mathbb{R}^{n \times k}$. In the clipping step, there are two cases for all $i \in [n]$.

**Case 1.** $\|\widetilde{Y}_{1,i}\|_2 \geq \xi$. We know

$$
\begin{aligned}
\|\widetilde{Y}_{1,i}Q\|_2 &= \|\widetilde{Y}_{1,i}\|_2 \\
&\geq \xi \\
&= \frac{2\mu k}{n},
\end{aligned} \tag{43}
$$

where the first step follows from Fact A.7, the second step follows from our Case 1 assumption, and the last step follows from the definition of $\xi$ (see Definition D.2).

We have

$$
\begin{aligned}
\|\widetilde{Y}_{1,i}Q - V_i\|_2 &\geq \|\widetilde{Y}_{1,i}Q\|_2 - \|V_i\|_2 \\
&\geq \frac{2\mu k}{n} - \|V_i\|_2 \\
&= \frac{2\mu k}{n} - \frac{\mu k}{n} \\
&= \frac{\mu k}{n},
\end{aligned} \tag{44}
$$

where the first step follows from the triangle inequality, the second step follows from Eq. (43), the third step follows from the property of $V_i$, and the last step follows from simple algebra.

By the definition of clipping, in this time, $\overline{Y}_{1,i} = 0$.

$$
\begin{aligned}
\|\overline{Y}_{1,i}Q - V_i\|_2 &= \|0 - V_i\|_2 \\
&= \|V_i\|_2 \\
&= \frac{\mu k}{n} \\
&\leq \|Q^\top \widetilde{Y}_{1,i} - V_i\|_2
\end{aligned}
$$

where the first step follows from $\overline{Y}_{1,i} = 0$ and the second step follows from simple algebra, the third step follows from the property of $V_i$, and the last step follows from Eq. (44).

**Case 2.** $\|\widetilde{Y}_{1,i}\|_2 < \xi$. In this case, we know

$$
\overline{Y}_{1,i} = \widetilde{Y}_{1,i}.
$$

Thus,

$$
\|\overline{Y}_{1,i}Q - V_i\|_2 = \|\widetilde{Y}_{1,i}Q - V_i\|_2.
$$

Combining Case 1 and Case 2, we know that for all $i \in [n]$,

$$
\|\overline{Y}_{1,i} - V_i\|_2 \leq \|\widetilde{Y}_{1,i} - V_i\|_2.
$$

Taking the summation of squares, we have

$$
\begin{aligned}
\|\overline{Y}_1 Q - V\|_F^2 &\leq \|\widetilde{Y}_1 Q - V\|_F^2 \\
&\leq \frac{1}{100}
\end{aligned} \tag{45}
$$

the last step follows from Eq. (42).

Eventually, we would like to show $V$ is close to $Y_1$. Suppose

$$
Y_1 = \overline{Y_1}R^{-1},
$$

where $R$ is an upper-triangular matrix.

Then,

$$
\begin{aligned}
\sin\theta(V, Y_1) &= \|V_\perp^\top Y_1\| \\
&= \|Y_1\|
\end{aligned}
$$

$$
\begin{aligned}
&= \|(\overline{Y}_1 - VQ^{-1})R^{-1}\| \\
&\leq \|\overline{Y}_1 Q - V\| \cdot \|R^{-1}\| \\
&\leq \|\overline{Y}_1 Q - V\| \cdot \frac{1}{\sigma_{\min}(\overline{Y}_1)} \\
&\leq \|\overline{Y}_1 Q - V\|_F \cdot \frac{1}{\sigma_{\min}(\overline{Y}_1)},
\end{aligned}
$$

where the first step follows from definition of $\sin$, the second step follows from Fact A.7, the third step follows from definition of $Y_1$, and the fourth step follows from $\|AB\| \leq \|A\| \cdot \|B\|$, the fifth step follows from singular values of $R$ and those of $\overline{Y}_1$ are identical, and the last step follows from $\|\cdot\| \leq \|\cdot\|_F$.

Note that

$$
\begin{aligned}
\sigma_{\min}(\overline{Y}_1) &= \sigma_{\min}(\overline{Y}_1 Q) \\
&\geq \sigma_{\min}(V) - \|\overline{Y}_1 Q - V\| \\
&\geq \sigma_{\min}(V) - \|\overline{Y}_1 Q - V\|_F \\
&\geq \sigma_{\min}(V) - 1/10 \\
&\geq \frac{1}{2},
\end{aligned}
\tag{46}
$$

where the first step follows from Fact A.7, the second step follows from Fact A.7, the third step follows from $\|\cdot\| \leq \|\cdot\|_F$, the fourth step follows from Eq. (45), and the last step follows from $\sigma_{\min}(V) = 1$.

Thus,

$$
\begin{aligned}
\sin\theta(V, Y_1) &\leq 2 \cdot \|\overline{Y}_1 Q - V\|_F \\
&\leq 2 \cdot \frac{1}{10} \\
&\leq \frac{1}{2},
\end{aligned}
\tag{47}
$$

where the first step follows from Eq. (46), and the second step follows from Eq. (45), the last step follows from simple algebra.

Therefore,

$$
\begin{aligned}
\mathrm{dist}(V, Y_1) &\leq 2\tan\theta(V, Y_1) \\
&\leq 4\sin\theta(V, Y_1) \\
&\leq 1/2,
\end{aligned}
$$

where the first step follows from Lemma A.14, the second step follows from Lemma A.14, the third step follows from Eq. (47).

**Proof of Part 2.** For $\rho(Y_1)$, we observe that

$$
Y_{1,i} = \overline{Y}_i R^{-1},
$$

We have

$$
\begin{aligned}
\|Y_{1,i}\|_2 &\leq \|\overline{Y}_{1,i}\|_2 \cdot \|R^{-1}\| \\
&\leq \xi \cdot \|R^{-1}\| \\
&\leq \xi \cdot \sigma_{\min}(\overline{Y}_1)^{-1} \\
&\leq \xi \cdot 2
\end{aligned}
\tag{48}
$$

where the first step follows from $\|Ax\|_2 \leq \|A\| \cdot \|x\|_2$, the second step follows from $\|\overline{Y}_{1,i}\|_2 \leq \xi$, and the third step follows from $\|R^{-1}\| = \sigma_{\min}(\overline{Y})^{-1}$, and last step follows from $\sigma_{\min}(\overline{Y}_1)^{-1} \leq 2$ (see Eq. (46)).

Note that

$$
\begin{aligned}
\rho(Y_{1,i}) &= \frac{n}{k} \cdot \max_{i \in [n]} \|Y_{1,i}\|_2 \\
&\leq \frac{n}{k} \cdot 2\xi \\
&\leq 4\mu
\end{aligned}
$$

where the first step follows from Definition A.8, the second step follows from Eq. (48), last step follows from $\xi = \mu k/n$. This leads to the bound, which completes the proof. $\quad\square$

### G.5 Main Result

Finally, in this section, we present our main results.

Table 3: Summary of our results.

| References | Init | Time |
|---|---|---|
| (Li et al., 2016) | Random | $\widetilde{O}((\|W\|_0 k^2 + nk^3) \log(1/\epsilon))$ |
| Theorem H.2 | Random | $\widetilde{O}((\|W\|_0 k + nk^3) \log(1/\epsilon))$ |
| (Li et al., 2016) | SVD | $O(n^3) + \widetilde{O}((\|W\|_0 k^2 + nk^3) \log(1/\epsilon))$ |
| Theorem G.7 | SVD | $O(n^3) + \widetilde{O}((\|W\|_0 k + nk^3) \log(1/\epsilon))$ |

**Theorem G.7** (Main result, SVD initialization). *Let $\eta = 1$. There is an algorithm starts from SVD initialization runs in $\log(1/\epsilon)$ iterations and generates $\widetilde{M}$, which is a matrix in $\mathbb{R}^{n \times n}$ and*

$$
\|\widetilde{M} - M^*\| \leq O(\alpha^{-1} \eta k \tau) \|W \circ N\|_F + \epsilon
$$

$$
O(n^3) + \widetilde{O}((\|W\|_0 k + nk^3) \log(1/\epsilon))
$$

*is the total running time.*

*Proof.* It follows directly from Lemma J.1 and Lemma J.2. $\quad\square$

## H Random Initialization

In Section H.1, we present our random initialization algorithm (see Algorithm 5) and analyze its properties. In Section H.2, we summarize our main result.

### H.1 Initialization

Now, we start to present Algorithm 5.

---
**Algorithm 5** Random Initialization
---
1: **procedure** RANDOMINIT($n, k$)
2: $\quad$ Let $Y \in \mathbb{R}^{n \times k}$ generated with $Y_{i,j} \leftarrow \frac{1}{\sqrt{n}} b_{i,j}$, where $b_{i,j}$ is drawn uniformly from $\{-1, +1\}$
3: $\quad$ **return** $Y$
4: **end procedure**

---

The following lemma shows that the minimum singular value of the matrix $Y^\top D_{W_i} Y$ can be lower bounded with high probability over all $i \in [n]$.

**Lemma H.1** (Random initialization, Lemma 9 in Li et al. (2016)). *Let $Y \in \mathbb{R}^{n \times k}$ be a random matrix with $Y_{i,j} = \frac{1}{\sqrt{n}} b_{i,j}$, where $b_{i,j}$ are independent and uniform variables from $\{-1, 1\}$. Let $\mu \geq 1$. Let $k \geq 1$. Let $\alpha > 0$. We assume that $\|W\|_\infty \leq \frac{\alpha}{k^2 \mu \log^2 n} \cdot n$. Then, we have*

$$
\Pr\left[\sigma_{\min}(Y^\top D_{W_i} Y) \geq \frac{\alpha}{4\mu k}, \ \forall i \in [n]\right] \geq 1 - 1/n^2.
$$

*Proof.* Notice that

$$Y^\top D_{W_i} Y = \sum_{j \in [n]} (Y_j)^\top (D_{W_i})_j Y_j.$$

For every $j \in [n]$,

$$(Y_j)^\top (D_{W_i})_j Y_j$$

is independent, and

$$\mathbb{E}[(Y_j)^\top (D_{W_i})_j (Y_j)] = \frac{1}{n} (D_{W_i})_j.$$

Using linearity of expectation, we have

$$\mathbb{E}[\sum_{j \in [n]} (Y_j)^\top (D_{W_i})_j (Y_j)] = \frac{1}{n} \sum_{j \in [n]} (D_{W_i})_j.$$

Then, we can get that the following equation holds. We use this first and will prove it from Eq. (50)

$$\sum_{j \in [n]} (D_{W_i})_j \geq \frac{\alpha n}{k \mu}. \tag{49}$$

Indeed, by the assumption weight is not degenerate, we can get that for all vectors $a$ in $\mathbb{R}^n$,

$$\begin{aligned}
a^\top V^\top D_{W_i} V a &= \sum_{j \in [n]} (D_{W_i})_j \langle V_j, a \rangle^2 \\
&\geq \min_{j \in [n]} \{ \langle V_j, a \rangle^2 \} \sum_{j \in [n]} (D_{W_i})_j \\
&= \frac{\mu k}{n} \sum_{j \in [n]} (D_{W_i})_j \\
&\geq \frac{\mu k}{n} \frac{\alpha n}{k \mu} \\
&= \alpha,
\end{aligned}$$

where the second step follows from Fact A.6, the third step follows from the incoherence of $V$, the fourth step follows from our claim (see Eq. (49)), and the last step follows from simple algebra.

Then, by the incoherence of $V$, we have

$$\sum_{j \in [n]} (D_{W_i})_j \langle V_j, a \rangle^2 \leq \sum_{j \in [n]} (D_{W_i})_j \frac{\mu k}{n}. \tag{50}$$

Hence,

$$\sum_{j \in [n]} (D_{W_i})_j \geq \frac{\alpha n}{k \mu}.$$

Combining everything together, we get

$$\mathbb{E}[\sum_{j \in [n]} (Y_j)^\top (D_{W_i})_j Y_j] \geq \frac{\alpha}{k \mu}.$$

Define

$$B := \|(Y_j)^\top (D_{W_i})_j Y_j\| \leq \frac{k}{n} (D_{W_i})_j$$

$$\leq \frac{\alpha}{k\mu \log^2 n},$$

where the first step follows from our sampling procedure and the second step follows from the assumption that $\|W\|_\infty \leq \frac{\alpha n}{k^2 \mu \log^2 n}$.

Since all the random variables

$$(Y_j)^\top (D_{W_i})_j Y_j$$

are independent, applying Matrix Chernoff we get that

$$\Pr[\sum_{j\in[n]} (Y_j)^\top (D_{W_i})_j Y_j \leq (1-\delta)\frac{\alpha}{k\mu}] \leq n(\frac{e^{-\delta}}{(1-\delta)^{(1-\delta)}})^{\frac{\alpha}{k\mu B}}$$

$$\leq n(\frac{e^{-\delta}}{(1-\delta)^{(1-\delta)}})^{\log^2 n}.$$

Picking $\delta = \frac{3}{4}$, and union bounding over all $i$, with probability at least $1 - \frac{1}{n^2}$, for all $i$,

$$\sigma_{\min}(Y^\top D_{W_i} Y) \geq \frac{\alpha}{4k\mu}$$

as needed.

$\square$

## H.2 MAIN RESULT

In this section, we summarize our main result.

**Theorem H.2** (Main result, random initialization). *Let $\eta$ be defined as Definition D.3. There is an algorithm starts from random initialization runs in $\log(1/\epsilon)$ iterations and generates $\widetilde{M}$, which is a matrix in $\mathbb{R}^{n\times n}$ and*

$$\|\widetilde{M} - M^*\| \leq O(\alpha^{-1}\eta k\tau)\|W \circ N\| + \epsilon.$$

$$\widetilde{O}((\|W\|_0 k + nk^3)\log(1/\epsilon)).$$

is the total running time.

*Proof.* We use Algorithm 5 to initialize $Y$. Then, we can use the proof of Lemma D.9, and $T$ is changed to

$$T = \{i \in [n] \mid \sigma_{\min}(Y^\top D_i Y) \leq 0.25\alpha/\eta\}.$$

where $\eta = \mu k$.

However, because of this change, $T = \emptyset$, with high probability. Then, the same calculation follows as in Lemma D.9. Note that in this case, Lemma F.1 is not needed because $S_1 = \emptyset$. Then, we can use Lemma J.1 and Lemma J.2, directly. $\square$

## I  BOUNDING THE FINAL ERROR

In Section I.1, we express $\widetilde{M} - M^*$ as a simpler form that is easier for further analysis. In Section I.2, we prove that $\|\widetilde{M} - M^*\|$ is bounded. Both of these are used to support the proof of our main theorem.

## I.1 REWRITE $\widetilde{M} - M^*$

In this section, we start to simplify/rewrite $\widetilde{M} - M^*$.

**Claim I.1.** *Let $M^* \in \mathbb{R}^{n \times n}$ is $(\mu, \tau)$-incoherent (see Definition 1). Let $\widetilde{M}$ be defined as Theorem 4.6. Then, we have*

$$\widetilde{M} - M^* = U\Sigma Q \Delta_y^\top + U\Sigma V^\top \Delta_y Y^\top + RY^\top$$

*Proof.* We start expanding the difference by definition:

$$
\begin{aligned}
&\widetilde{M} - M^* \\
&= \overline{X}Y^\top - M^* \\
&= (U\Sigma V^\top (VQ + \Delta_y) + R)(VQ + \Delta_y)^\top - U\Sigma V^\top \\
&= U\Sigma V^\top (VQ + \Delta_y)(VQ + \Delta_y)^\top + R(VQ + \Delta_y)^\top \\
&\quad - U\Sigma V^\top \\
&= U\Sigma V^\top VQ(VQ + \Delta_y)^\top + U\Sigma V^\top \Delta_y (VQ + \Delta_y)^\top + R(VQ + \Delta_y)^\top - U\Sigma V^\top \\
&= U\Sigma V^\top VQQ^\top V^\top - U\Sigma V^\top + U\Sigma V^\top VQ\Delta_y^\top + U\Sigma V^\top \Delta_y (VQ + \Delta_y)^\top + R(VQ + \Delta_y)^\top \\
&= U\Sigma V^\top VQ\Delta_y^\top + U\Sigma V^\top \Delta_y (VQ + \Delta_y)^\top + R(VQ + \Delta_y)^\top \\
&= U\Sigma Q \Delta_y^\top + U\Sigma V^\top \Delta_y Y^\top + RY^\top,
\end{aligned}
$$

where the first step follows from $\widetilde{M} = \overline{X}Y^\top$, the second step follows from $\overline{X} = U\Sigma V^\top Y + R$, $Y = VQ + \Delta_y$, and $M^* = U\Sigma V$, the third step follows from simple algebra, the fourth step follows from the simple algebra, the fifth step follows from that for all matrices $A, B$, $(A + B)^\top = A^\top + B^\top$, the sixth step follows from $V \in \mathbb{R}^{n \times k}$ is an orthogonal matrix and $Q \in \mathbb{R}^{k \times k}$ is a rotation matrix, and the last step follows from $Y = VQ + \Delta_y$. $\square$

## I.2 BOUNDING $\|\widetilde{M} - M^*\|$

In this section, we bound $\|\widetilde{M} - M^*\|$.

**Claim I.2.** *Let $M^* \in \mathbb{R}^{n \times n}$ is $(\mu, \tau)$-incoherent (see Definition 1). Let $\widetilde{M}$ be defined as Theorem 4.6. Then, we have*

$$\|\widetilde{M} - M^*\| \leq (2\Delta_F)\|W \circ N\| + \epsilon$$

*Proof.*

$$
\begin{aligned}
\|\widetilde{M} - M^*\| &= \|U\Sigma Q \Delta_y^\top + U\Sigma V^\top \Delta_y Y^\top + RY^\top\| \\
&\leq \|U\Sigma\|\|Q\|\|\Delta_y\| + \|U\Sigma V^\top\|\|\Delta_y\|\|Y\| + \|R\|\|Y\| \\
&\leq \Theta(1) \cdot \|\Delta_y\| + \|R\| \\
&\leq O(\mathrm{dist}(Y, V)) + \Delta_f \cdot \|W \circ N\| \\
&\leq \frac{1}{2^t} + \Delta_F \cdot \|W \circ N\| + \Delta_f \cdot \|W \circ N\| \\
&= (\Delta_F + \Delta_f)\|W \circ N\| + \frac{1}{2^t} \\
&\leq (2\Delta_F)\|W \circ N\| + \frac{1}{2^t} \\
&\leq (2\Delta_F)\|W \circ N\| + \epsilon
\end{aligned}
$$

where the second step is due to the inequalities $\|A + B\| \leq \|A\| + \|B\|$ and $\|AB\| \leq \|A\|\|B\|$, the third step is supported by $\|U\Sigma\| = \|\Sigma\|$, $\|Q\| = 1$, $\|U\Sigma V^\top\| = \|\Sigma\| = 1$ (See Definition 1), $\|Y\| = 1$, the second last step follows from $\Delta_f \geq \Delta_F$, and the last step follows from $t = O(\log(1/\epsilon))$. $\square$

## J  PROOF OF MAIN RESULT

We dedicate this section to the proof of Theorem 4.6. There are two parts of Theorem 4.6: the correctness part and the running time part. In Section J.1, we present the proof of the correctness part of Theorem 4.6. In Section J.2, we display the proof of the running time part of Theorem 4.6.

### J.1  CORRECTNESS PART OF THEOREM 4.6

Now, we start proving the correctness part of Theorem 4.6.

**Lemma J.1** (Correctness part of Theorem 4.6). *Suppose $M^* \in \mathbb{R}^{n \times n}$ is $\mu$-incoherent (see Assumption 1). Assume that $W$ has $\gamma$-spectral gap (see Assumption 2) and $(\alpha, \beta)$-bounded (see Assumption 3). Let $\gamma$ satisfy condition in Definition D.4.*

*There is an algorithm (Algorithm 1) takes $M^* + N \in \mathbb{R}^{n \times n}$ as input, uses either SVD initialization or random initialization and runs in $O(\log(1/\epsilon))$ iterations and generates $\widetilde{M}$, which is a matrix in $\mathbb{R}^{n \times n}$ and*

$$\|\widetilde{M} - M^*\| \leq O(\alpha^{-1} k \tau) \cdot \|W \circ N\| + \epsilon.$$

*Proof.* By Lemma E.1, we can prove, for any $t > 1$

$$\mathrm{dist}(X_t, U) \leq \frac{1}{2^t} + 100\alpha^{-1}\sigma_{\min}(M^*)^{-1}k \cdot \|W \circ N\|,$$

$$\mathrm{dist}(Y_t, V) \leq \frac{1}{2^t} + 100\alpha^{-1}\sigma_{\min}(M^*)^{-1}k \cdot \|W \circ N\|. \tag{51}$$

Note that $\sigma_{\min}(M^*)^{-1} \leq \tau$ (see Definition 1), then the above statement becomes

$$\mathrm{dist}(X_t, U) \leq \frac{1}{2^t} + \Delta_F \cdot \|W \circ N\|,$$

$$\mathrm{dist}(Y_t, V) \leq \frac{1}{2^t} + \Delta_F \cdot \|W \circ N\|.$$

where $\Delta_F := 5\Delta_f$. By Lemma D.9 and Claim D.7,

$$\|\overline{X} - U\Sigma V^\top Y\|_F$$
$$\leq \Delta_d \cdot \mathrm{dist}(Y, V) + \Delta_f \cdot \|W \circ N\|, \tag{52}$$

where $\Delta_d$ and $\Delta_f$ are defied as Definition D.5.

To promise the first term in $\Delta_d^2$ is less than $0.1$ and using Lemma A.3, we need to choose (note that $c_0$ is defined as Definition D.4)

$$\gamma \leq \frac{1}{20} \cdot \frac{\alpha}{\mathrm{poly}(\mu, k) \cdot n^{c_0}}$$

To promise the second term in $\Delta_d^2$ is less than $0.1$, we have to choose

$$\gamma \leq \frac{1}{20} \cdot \frac{\alpha}{\mathrm{poly}(\mu, k, \tau)}$$

For $\mathrm{dist}(Y, V)$, let

$$P := \arg\min_{Q \in O^{k \times k}} \|YQ - V\|.$$

We define

$$V := YP + \Delta$$

and $Y = VP^\top - \Delta P^\top$.

Let $Q := P^\top \in O^{k \times k}$ and $\Delta_y := -\Delta P^\top$, then

$$Y = VQ + \Delta_y$$

with $\|\Delta_y\| = \mathrm{dist}(Y, V)$.

We define

$$R := \overline{X} - U\Sigma V^\top Y.$$

Then Eq. (52) implies that

$$\|R\|_F \le \mathrm{dist}(Y, V) + \Delta_f \|W \circ N\|$$

Let $\overline{X} := \overline{X}_{T+1}$ and $Y := Y_T$, then

$$\widetilde{M} = \overline{X} Y^\top.$$

Using Claim I.1, we have

$$\widetilde{M} - M^* = U\Sigma Q \Delta_y^\top + U\Sigma V^\top \Delta_y Y^\top + R Y^\top. \tag{53}$$

Using Claim I.2, we have

$$\|\widetilde{M} - M^*\| \le (2\Delta_F)\|W \circ N\| + \epsilon. \tag{54}$$

$\square$

## J.2 RUNNING TIME PART OF THEOREM 4.6

Now, we start proving the running time part of Theorem 4.6.

**Lemma J.2** (Running Time Part of Theorem 4.6). *The running time of Algorithm 1 is $\widetilde{O}((\|W\|_0 k + nk^3)\log(1/\epsilon))$ with random initialization.*

*Proof.* Now we analyze the running time. We first compute the initialization time. The entry $Y_{i,j}$ of the matrix $Y$ is equal to $\frac{1}{\sqrt{n}} b_{i,j}$, where $b_{i,j}$'s are independent uniform from $\{-1, 1\}$. Hence, the time complexity of random initialization is $O(nk)$. There are $T$ iterations. For each iteration, there are three major steps, solving regression, Clip and QR. The dominating step is to solve regression. We choose $\epsilon_{\mathrm{sk}}$ as Claim D.7. Using Lemma C.2 and Lemma C.3, we know that we should choose $\epsilon_0 = \epsilon_{\mathrm{sk}}/\mathrm{poly}(n)$ and $\delta_0 = 1/\mathrm{poly}(n, \log(1/\epsilon))$, this step takes $\widetilde{O}(\|W\|_0 k + nk^3)$ time. The CLIP and QR algorithms take time $O(nk)$ and $O(nk^2)$ respectively. Hence, the $T$ iterations take time $\widetilde{O}((\|W\|_0 k + nk^3)\log(1/\epsilon))$. $\square$

# K EXPERIMENTAL RESULTS

We conducted two experiments showing the performance of our main algorithm (Algorithm 1) and one experiment, particularly for our novel high-precision regression algorithm (Algorithm 2). We first present the first two experiments for our main algorithm. In both experiments, we set

$$M = M^* + N \in \mathbb{R}^{n \times n}$$

where $M^*$ is the rank-$k$ ground truth and $N$ is a higher-rank noise matrix. We set $n = 800$ and $k = 100$. We generate the noise matrix $N$ as an $n \times n$ random matrix with i.i.d. Gaussian entries of zero mean and variance $\frac{1}{k}$. We apply the sketching matrix $S \in \mathbb{R}^{m \times n}$ with $m = 150$ (we choose the CountSketch matrix (Charikar et al., 2002) when solving for the regression problems (see Lines 7 and 10 from our Algorithm 1). We iterate the alternating minimization steps (see Line 6) for $T = 20$ times. To show the performance of our algorithm, we compare the running time and error between our Algorithm 1 and the exact solver from Li et al. (2016).

**Experiment 1** The first experiment is the matrix completion problem. For each row of the weight matrix $W \in \mathbb{R}_{\ge 0}^{n \times n}$, we randomly select 400 entries to be equal to 1 and the remaining 400 entries to be 0. The second experiment is the general weighted low rank approximation where the weight matrix $W$ is constructed via $\mathbf{1}_n \mathbf{1}_n^\top + G$ for $G$ being a random matrix with standard Gaussian entries.

Below, we present our experimental results of the matrix completion problem. We generate the ground truth $M^* = XY^\top$ by a pair $X, Y \in \mathbb{R}^{n \times k}$ with random i.i.d. entries scaled by $\frac{1}{\sqrt{k}}$, and the distributions are Laplace, Gaussian, and uniform, respectively. Our time is measured in seconds.

Table 4: Experimental results of Algorithm 1 for the matrix completion problem.

| Distribution | Time of the exact solver | Time of the approximate solver | % speedup | Exact solver error | Approximate solver error |
|---|---|---|---|---|---|
| Laplace | 234 | **205** | 12.39% | $1.54 \times 10^{-4}$ | $3.56 \times 10^{-3}$ |
| Gaussian | 247 | **216** | 12.55% | $3.85 \times 10^{-4}$ | $1.33 \times 10^{-3}$ |
| Uniform | 248 | **207** | 16.53% | $1.34 \times 10^{-5}$ | $4.99 \times 10^{-4}$ |

Table 5: Experimental results of Algorithm 1 for the weighted low-rank approximation.

| Distribution | Time of the exact solver | Time of the approximate solver | % speedup | Exact solver error | Approximate solver error |
|---|---|---|---|---|---|
| Laplace | 249 | **222** | 10.84% | $2.11 \times 10^{-4}$ | $4.71 \times 10^{-4}$ |
| Gaussian | 223 | **204** | 8.52% | $6.29 \times 10^{-5}$ | $2.15 \times 10^{-4}$ |
| Uniform | 221 | **215** | 2.71% | $3.00 \times 10^{-5}$ | $7.13 \times 10^{-5}$ |

**Experiment 2** We present our second experimental results as follows (recall the weight matrix is an all-1's matrix plus a noise matrix with i.i.d. standard Gaussian entries):

We note that in these two settings, our algorithm achieves a speedup compared to the algorithm of Li et al. (2016). For the matrix completion setting, our speedup is in the range of 12%-16%, while for the dense weights regime, our speedup is roughly 10%. In 20 iterations, our approximate solver obtains errors similar to those of the exact solver.

**Experiment 3** We expect the speedup will be more significant once the discrepancy between $n$ and $k$ is larger, as sketching is known to work well in the regime where $n \gg k$, as evidenced by the following experiment on using sketching to solve the regression:

$$\min_{x \in \mathbb{R}^k} \|Ax - b\|_2^2 \quad \text{for } A \in \mathbb{R}^{n \times k} \text{ and } b \in \mathbb{R}^n.$$

We test the performance of regression solvers for $n = 10^6$, $k = 500$ with a sketch size $m = 5500$, and run our solver for 5 iterations. The results are as follows:

Table 6: Experimental results of Algorithm 2.

| Distribution | Time for exact solve | Time for approx solve | Percentage speedup | Error |
|---|---|---|---|---|
| Gaussian | 5.149 | **3.695** | 28.24% | $8.24 \times 10^{-3}$ |
| Laplace | 5.480 | **3.866** | 29.45% | $8.41 \times 10^{-3}$ |
| Power Law | 5.505 | **4.444** | 19.27% | $8.77 \times 10^{-3}$ |

Our data matrices $A$ and response vectors $b$ are generated according to standard Gaussian, Laplace, and power law distribution with $p = 5$, and we measure the $\ell_\infty$ error of the solution, i.e., let $\widehat{x}$ denote the vector outputted by the solver, we measure

$$\|\widehat{x} - x^*\|_\infty$$

where

$$x^* = \arg \min_{x \in \mathbb{R}^k} \|Ax - b\|_2^2.$$

The speedup obtained here ranges from 20% to 30%, so we have strong grounds to believe that the acceleration will be even more evident when $n$ is large. However, performing weighted low-rank approximation on $10^6 \times 10^6$ size matrices is currently out of the scope of our computational power, so we leave this as a future direction.

