# OpenReview forum: "Efficient Alternating Minimization with Applications to Weighted Low Rank Approximation"
_ICLR.cc/2025/Conference — ICLR 2025 Poster_

### Official Review · Reviewer_5iC9 · 2024-10-31

**Soundness:** 3
**Presentation:** 3
**Contribution:** 3
**Rating:** 6
**Confidence:** 4

**Summary:**

In this work, the authors proposed a new alternating minimization method for the weighted low-rank matrix approximation problem. They utilized an efficient multiple-response regression solver to reduce the computational time from ${O}((\|W\|_0k^2 + nk^3)\log(1/\epsilon))$ to $\tilde{O}((\|W\|_0k + nk^3)\log(1/\epsilon))$.

**Strengths:**

The paper provides a novel algorithm for the low-rank matrix approximation problem. The results in this work may also be applied to other optimization and machine learning problems. After checking the paper, I think the theoretical results in the paper should be sound, although I did not check all proofs due to the time limit.

**Weaknesses:**

Although this paper is mostly a theoretical paper, I think it is still beneficial to include numerical illustrations to verify the theoretical findings and show the usefulness of the proposed method. For example, the authors may generate artificial datasets that satisfy the assumptions in this work and compare their method with the methods in (Li et al., 2016). It would be even better if the authors can verify that Assumption 2.3 holds in some practical examples and exhibit the empirical performance on these real-world examples.

Moreover, in Line 223, the authors claimed that Assumption 3 is a generalization to the RIP condition. Although I agree with the authors that the assumption is more general in the sense that it contains the weight matrix $W$, I think it is more restrictive than the RIP condition. Suppose that $W$ is a boolean matrix and each column of $W$ contains at most $k$ 1's. In this case, the assumption considers at most $n$ combinations of rows of $U$ and $V$. In contrast, the RIP condition requires the error bound to hold for all $k$-dimensional subspace. In addition, I feel that Assumption 3 is not directly comparable with the classical RIP condition. The definition of the classic $(r, \delta)$-RIP condition would imply that $(1-\delta) \lVert K \rVert_F^{2} \leq \lVert W \circ K \rVert_F^{2} \leq (1+\delta) \lVert K \rVert_{F}^{2}$ holds for any matrix $K$ of rank at most $k$, which is independent of $M^*$. I would suggest the authors include more detailed explanations to this claim.

**Questions:**

I have a few other minor comments for the authors to consider:

- Theorem 4.6: I wonder if the time complexity also depends on the scale of $M^*$ and $N$. For example, let $c>0$ be a constant. If we multiply $M^*$ and $N$ by $c$, then the solution $\tilde{M}$ will also be multiplied by $c$. However, the value $\epsilon$ is not changed in the error bound. In addition, Assumption 2.3 does not bound the scale of $M^*$ and $N$ either. If I did not miss anything, I feel that the number of iterations or the error $\epsilon$ should depend on the scale of $M^*$ and $N$.


- Line 499: it would be helpful if the authors could provide supporting evidence for the claim that the spectral norm is more robust than the Frobenius norm.

---

> ### Author Response · Authors · 2024-11-22
>
> We express our deepest gratitude for your insightful comments and sincerely appreciate your evaluation of our work as technically sound, with techniques that can be applied to other optimization and machine learning problems. Your feedback is extremely valuable to us and reinforces our belief that our work will have a meaningful impact on the optimization and machine learning communities. Below, we address your concerns about our work.
>
> **Concern**: It is beneficial to include numerical illustrations to verify the theoretical findings and show the usefulness of the proposed method.
>
> **Answer**: Regarding the first weakness, we kindly refer you to our global response. We have conducted preliminary experiments demonstrating that our algorithm is indeed faster than the exact solver proposed in (Li et al., 2016). We believe that our experimental results strongly suggest that our algorithm will perform effectively in practice.
>
> **Concern**: In Line 223, the paper shows that Assumption 3 is a generalization to the RIP condition. Although the reviewer agrees that the assumption is more general in the sense that it contains the weight matrix $W$, the reviewer thinks it is more restrictive than the RIP condition.
>
> **Answer**: We agree with your observation that our wording is imprecise in referring to it as generalized RIP. For a matrix $A$ to be RIP, one requires for any subset of columns of size $s$, we have $(1-\delta_s) I_s \preceq A_s^\top A_s\leq (1+\delta_s) I_s$, but here, we only have that $\alpha I_k \preceq U^\top D_{W_i} U \preceq \beta I_k$ for all $n$ columns of $W$. This should instead be treated as a generalization of Assumption A2 in (Bhojanapalli et al., 2014), where they consider the noisy matrix completion problem and $D_{W_i}$ is a binary diagonal matrix. A more precise description of this assumption is that it represents a stronger condition than incoherence, which is itself implied by the stronger incoherence assumption in (Candès et al., 2010).
>
> **Question**: Theorem 4.6: I wonder if the time complexity also depends on the scale of $M^*$ and $N$. For example, let $c > 0$ be a constant. If we multiply $M^*$ and $N$ by $c$, then the solution $\tilde{M}$ will also be multiplied by $c$. However, the value $\epsilon$ is not changed in the error bound. In addition, Assumption 2.3 does not bound the scale of $M^*$ and $N$ either. If I did not miss anything, I feel that the number of iterations or the error $\epsilon$ should depend on the scale of $M^*$ and $N$.
>
> **Answer**: Our runtime actually does not depend on the scale of $M$ and $M^*$. Intuitively, this is because the scaling is accounted for by the alternating minimization: if we scale $M$ by a factor of $c>0$, then the singular vectors remain unchanged, and singular values are scaled by $c$. For simplicity, assuming in Algorithm 1, we do not need to perform Clip, and instead of randomly initializing $Y$, we set $Y$ to be the right singular vectors of $W\circ M$. Note that in the iteration, when solving for $X$, the magnitude of $X$ will account for the scaling of $M$, i.e., compared to the original $X$ without scaling by $c$, the new $X$ will be scaled by a factor of $C$. After properly renormalizing $X$, the new $Y$ will also account for the scaling of $M$. If one inspects our proof, we essentially prove that the distance between the subspace spanned by the optimal right singular vectors of $M^*$ and the initial $Y$ is at most $\frac{1}{2}$, and we inductively prove that the distance shrinks geometrically. The distance between subspaces is invariant under scaling as they are defined in terms of the orthonormal basis for the subspace, therefore the $\epsilon$ term is not affected by the scaling of $M$. Of course, in the most rigorous sense, it is not precise to state that our runtime “does not depend on the scale of $M$”, as we implicitly assume that operations such as matrix multiplication, inversion, and SVD can be done in $O(n^3)$ time, which is not true if the entries of $M$ are too large (exponentially large, which would require $\mathrm{poly}(n)$ bits to write down).

---

> > ### Author Response · Authors · 2024-11-22
> >
> > **Question**: Line 499: it would be helpful if the authors could provide supporting evidence for the claim that the spectral norm is more robust than the Frobenius norm.
> >
> > **Answer**: For the sake of argument we want to obtain a matrix whose Frobenius norm error to the ground truth is at most $\epsilon$. Using our spectral norm error algorithm, we could simply run our algorithm with the accuracy parameter $\epsilon/n$ and increase the runtime by a factor of $\log n$ to achieve this goal. Note that this yields a stronger guarantee than a mere $\epsilon$ error in the Frobenius norm, as by Weyl’s Theorem, we have that all singular values are approximately within a range of $\epsilon/n$. Intuitively, this means that our solution obtains a “flat” error spectrum: it gives a good approximation in all directions.
> > On the other hand, an algorithm that only achieves a Frobenius norm error bound loses the “flatness”: it could well be the case that there are a few singular values that are $O(\epsilon)$ away from the ground truth, while the remaining singular values are approximated extremely well. Hence, we argue that an algorithm with spectral norm error is more robust than one with Frobenius norm error.
> >
> > Once again, we deeply appreciate all the insightful comments you gave us, and we hope that our response may address your concerns. Please let us know if you have any further questions or concerns. We are more than willing to address any concerns or comments you may have.
> >
> > Yuanzhi Li, Yingyu Liang, and Andrej Risteski. "Recovery guarantee of weighted low-rank approximation via alternating minimization." ICML’16.
> >
> > Srinadh Bhojanapalli, and Prateek Jain. "Universal matrix completion." ICML’14.
> >
> > Emmanuel J. Candès, and Terence Tao. "The power of convex relaxation: Near-optimal matrix completion." IEEE transactions on information theory’10.

---

### Official Review · Reviewer_j2kz · 2024-10-31

**Soundness:** 3
**Presentation:** 2
**Contribution:** 3
**Rating:** 5
**Confidence:** 4

**Summary:**

This paper considers the weighted low-rank approximation problem, which can be solved using alternating minimization. In previous methods, the alternating minimization algorithm requires solving n linear regression problems each iteration. In this work, the authors develop a framework based sketching, and they prove that alternating minimization is robust enough such that the approximate solution return each iteration also converges towards the optimal solution. As a result, they obtain an improved running time compared to Li et al. 2016.

**Strengths:**

I think the theoretical results in this paper is solid: the proof that the alternating minimization framework is robust enough to tolerate large errors induced in the approximate solution is interesting, and can possibly be generalized to other similar problems in matrix optimization.

**Weaknesses:**

I see two main weaknesses. First, I think the writing of this paper needs to be improved, especially in section 3, where the algorithm is first introduced. Algorithm 1 is outlined in a dense way that makes it hard for the reader to parse exactly what is going on. Also, lines 270-280 contain numerous sentences that are a bit awkward or hard to understand. I suggest that the authors use a separate section to slowly present algorithm 1 step by step, and describe which parts are new.

Another weakness i see is the lack of any numerical results. In the introduction (lines 80) the authors say that the results of Li et al. 2016 are hard to deploy in practice because of the high time complexity. While this paper reduces the time complexity in theory, there are no numerical section to demonstrate that the reduction in time complexity is useful in practice. I suggest that the authors at least run some synthetic experiments to demonstrate a reduction in real runtime.

**Questions:**

Can the authors comment on the performance of their algorithm on some real or even synthetic problems?

---

> ### Author Response · Authors · 2024-11-22
>
> We express our deepest gratitude for your insightful comments! We truly appreciate your recognition that our theoretical results are solid, interesting, and have the potential to influence other theoretical areas. Below, we address your concerns about our work.
>
> **Concern**: The writing of the paper could be improved, for example, Algorithm 1 is outlined in a dense way in Section 3. Also, lines 270 - 180 contain numerous sentences that are a bit awkward and hard to understand.
>
> **Answer**: We note that we do not introduce any algorithm in Section 3. We assume you are referring to the bottom of Section 1, where we introduce Algorithm 1. Compared to the original work of (Li et al., 2016), the main innovation of our algorithm is a strong and robust analysis when the alternating updates $\vec X$ and $\vec Y$ are computed approximately and efficiently. We have marked these two lines in red to emphasize this distinction. In the revised version of our paper (see Remark 1.2 that we added), we explicitly state that the general framework of our main algorithm (Algorithm 1) is a strengthening of the traditional alternating minimization, by replacing the exact update with an approximate update (lines 7 and 10) to make the overall algorithm faster and more robust. We hope this clarification will help readers understand that while our main algorithm is lengthy, the key contribution lies in our approximate fast solver (Algorithms 2 and 3). And the majority of Section 33 is centered around these two topics: 1. How to design a faster solver for approximate alternating updates, 2. How to prove the convergence of the algorithm under the presence of errors introduced by approximate updates.
> Regarding Lines 270 to 280, thank you very much for pointing this out. There is a typo on Line 279 (which is now Line 281 in the updated version). The first word on that line should be “forward” instead of “backward.” In general, we intended to convey that, although the traditional sketching technique is fast, it has a limitation that makes it unsuitable for implementation in our algorithm. Specifically, it provides a relative error guarantee on the regression cost (forward error). While this forward error can be transformed into the desired backward error, it ultimately causes the backward error to blow up, which negatively impacts the running time of our algorithm.
>
> **Concern**: The paper lacks numerical experiments, I suggest that the authors at least run some synthetic experiments to demonstrate a reduction in real runtime.
>
> **Answer**: We kindly refer you to our global response. We have conducted preliminary experiments demonstrating that our algorithm is indeed faster than the exact solver proposed in (Li et al., 2016). We believe that our experimental results strongly indicate that our algorithm will perform well in practice.
>
> Again, we deeply appreciate all of your insightful comments. We have changed our manuscript accordingly to highlight the important parts of our Algorithm 1 and refer the readers to (Li et al., 2016) for more alternating minimization background to understand the remaining parts of our Algorithm 1 (as specified above), and we have fixed our typo, making Lines 270 to 280 more clear.
>
> Yuanzhi Li, Yingyu Liang, and Andrej Risteski. "Recovery guarantee of weighted low-rank approximation via alternating minimization." ICML’16.

---

### Official Review · Reviewer_MHu4 · 2024-11-02

**Soundness:** 3
**Presentation:** 3
**Contribution:** 3
**Rating:** 6
**Confidence:** 2

**Summary:**

The authors propose an efficient framework for alternating minimization, which allows the subproblems to be solved approximately. They apply the framework to the weighted low rank approximation problem, theoretically improving the runtime.

**Strengths:**

1. The authors propose a high-precision regression solver and extend it to the weighted case. They also demonstrate the robustness of the alternating minimization framework. The theoretical analysis is a significant contribution.
2. The motivation behind their method is detailed and well-presented in the paper.

**Weaknesses:**

The authors did not conduct experiments to verify the performance of their algorithm. I am interested in whether the algorithm is feasible in practice.

**Questions:**

1. Could the authors explain the connection between the problem they considered (Problem 2.4) and the original problem? Are they equivalent under some conditions?
2. Could the authors explain the meaning of $\epsilon_0$ in Line 351?

---

> ### Author Response · Authors · 2024-11-22
>
> We express our deepest gratitude to the reviewer for the time and effort in giving us insightful feedback, and thank you for evaluating our work as well-motivated, well-presented, and significantly contributing to theory. Below, we want to address your concerns about our work.
>
> **Concern**: The authors didn’t conduct experiments to verify the performance of their algorithm.
>
> **Answer**: We kindly refer you to our global response. We have conducted preliminary experiments demonstrating that our algorithm is indeed faster than the exact solver proposed in (Li et al., 2016). We believe that our experimental results strongly indicate that our algorithm is practical and effective.
>
>
> **Question**: Could the authors explain the connection between the problem they considered (Problem 2.4) and the original problem? Are they equivalent under some conditions?
>
> **Answer**: To see the connection, define $f(K):=\\| W \circ (M - K) \\|_F^2$ where $K$ is rank-$k$. The original weighted low rank approximation problem aims to find an approximate solution with small \emph{forward error}, i.e., finding a $\widetilde K$ such that $f$ is approximately minimized. Note that $f$ is non-convex, so even if we can find $\widetilde K$ where $f(\widetilde K)$ is small, it is not guaranteed that $\widetilde K$ is close to $M$ in any form. On the other hand, the goal defined in Problem 2.4 seeks to find a matrix that is not only close to $M$ in spectral norm but also assumes $M=M^*+N$ where $M^*$ is rank-$k$ ground truth and $N$ is a high-rank noise. The objective is to find a $\widetilde M$ that is close to the ground truth $M^*$. In other words, the guarantee we achieve in this paper is the \emph{backward error} guarantee. Note that $\widetilde M$ is close to $M^*$ already implies that $f(\widetilde M)$ is small. Therefore, the guarantee provided in this paper strengthens the original weighted low rank approximation problem. Our algorithm not only solves the original problem efficiently in $\\|W\\|_0 k$ time but also addresses a stronger and more challenging problem.
>
>
> **Question**: Could the authors explain the meaning of $\epsilon_0$ in Line 351?
>
> **Answer**: We appreciate you for pointing this out, the $\epsilon_0$ in Line 351 is in fact a typo, and it was supposed to be $\epsilon$. We have fixed this issue in the updated manuscript.
>
>
> We thank you again for your insightful comments. We hope that our comments may address your concerns. If you have any further questions or concerns about our work, we are more than happy to address them.
>
>
> Yuanzhi Li, Yingyu Liang, and Andrej Risteski. "Recovery guarantee of weighted low-rank approximation via alternating minimization." ICML’16.

---

> > ### Comment · Reviewer_MHu4 · 2024-11-22
> > **Response to Rebuttal**
> >
> > I appreciate the authors taking the time to respond to my questions. I appreciate the explanation of the connection between Problem 2.4 and the original problem, as well as the additional numerical experiments provided by the author. I am satisfied with the authors' rebuttal.

---

### Official Review · Reviewer_dS4q · 2024-11-04

**Soundness:** 3
**Presentation:** 3
**Contribution:** 3
**Rating:** 6
**Confidence:** 4

**Summary:**

This paper studies the problem of weighted low-rank matrix approximation. In this problem one is given a weight matrix W
 and observes a matrix M', and the goal is to recover a matrix M of rank at most k that minimizes  $\Vert W \circ (M'- M)\Vert$..

This paper provides new theoretical guarantees within the alternating minimization framework for weighted low-rank matrix approximation. The authors propose a new alternating minimization algorithm that uses high-accuracy regression solver based on sketching instead of exact regression solvers in the optimization step of the algorithm compared to the work of (Li et al, 2016). This allows authors to improve runtime by a factor of $k$.
.

**Strengths:**

The problem of weighted low-rank matrix approximation is an important practical problem and an important algorithmic primitive that arises in numerical linear algebra, and various machine learning applications. The problem naturally generalizes classical low-rank matrix approximation problems and a matrix completion problem. Alternating minimization is an extremely successful practical approach to matrix completion problem, so establishing theoretical guarantees for this approach to the weighted low-rank matrix approximation problem is of high interest.

This paper builds on top algorithm analyzed in (Li et al., 2016) and the key difference between the algorithm in (Li et al., 2016) and this paper is that exact solvers for linear regression is replaced with an approximate solver that can be traced back to (Rokhlin & Tygert, 2008). This allows authors to improve runtime by a factor of $k$ and to get a nearly optimal runtime.

**Weaknesses:**

The assumption 2.4.2 asks that $||W - J||<\gamma n$, where J is an all-one matrix with $\gamma = O(n^{-c})$ (compared to O(1) in  (Li et al., 2016)). This assumption sounds quite strong to me. This means that in the matrix completion scenario, this allows one to have only o(1)-fraction of entries not observed in every row. Considering that one of the key regimes for low-rank problems assumes that the rank satisfies k = O(1), I am not sure if improvement in runtime completely justifies slightly stronger assumption on W.

Another slight weakness of the paper is that most of the key techniques used in this paper were introduced in prior work, so the paper lacks a bit of novelty in this sense.

**Questions:**

Are you aware of any practical application where the improvement in the running time by a factor of k obtained in this paper may be significant to applications, and where one might expect assumptions to hold?

---

> ### Author Response · Authors · 2024-11-22
>
> We express our deepest gratitude to the reviewer for these insightful comments. We thank you for pointing out that the problem we study is important and the reduction in the running time over the top algorithm analyzed in (Li et al., 2016). Below, we want to address your concerns about our work.
>
> **Concern**: The assumption that $\\|W - J \\|\leq \gamma n$ with $\gamma=O(n^{-c})$ is quite strong compared to $\gamma=O(1)$ in (Li et al., 2016). Can you justify the slightly stronger assumption on $W$?
>
> **Answer**: We note that our result is best compared to the random initialization of (Li et al., 2016) (Theorem 3), where they additionally impose a bound on $\\|W \\|\_{1, \infty}\leq O(\frac{\alpha n}{k^2\mu\log n})$ (in their notation, $\\|W \\|\_{\infty}$). Note that $\\|W \\|\_{1, \infty}$ gives an upper bound on the spectral norm, thereby implicitly bounding $\\| W - J \\|$ and the choice of $\gamma$. To simplify their additional assumption, we choose $\gamma$ to be smaller, such that it directly implies the desired $\\|W\\|_{1, \infty}$ bound as in (Li et al., 2016). To obtain a similar bound for $\gamma$  that is roughly $O(\frac{1}{\mu^2 k^{2.5}})$, we could instead pose the same assumptions on $\\|W\\|\_{1, \infty}\leq O(\frac{\alpha n}{k^2 \mu \log n})$. Therefore, our result should be interpreted as a robustification and direct speedup of (Li et al., 2016) without introducing stronger assumptions or restricting parameter choices (up to polylogarithmic factors). In the appendix (see Theorem G.7), we prove a similar result as the SVD initialization of (Li et al., 2016) that does not need the bound on $\\|W\\|\_{1, \infty}$.
>
> **Concern**: The key techniques used in this paper were introduced in prior works, so it lacks novelty a bit in that sense.
>
> **Answer**: While we build on existing techniques, we would like to emphasize a major novel insight of our proofs: a significant simplification of the analysis for noiseless matrix completion introduced by (Gu et al., 2024). In that work, the authors develop a convoluted framework to accelerate noiseless matrix completion. Their analysis, in particular, is heavily built on the specific structure of that problem, making it unclear how to generalize their techniques to more complex problems, such as matrix completion with noise or weighted low rank approximation. In contrast, our analysis is both simpler and more robust, and we believe it could be extended to problems such as multi-view learning (though we leave this as a future direction). We hope our work provides insights into developing robust and efficient alternating minimization methods for a broader range of problems.
>
> **Question**: Are you aware of any practical application where the improvement in the running time by a factor of $k$ obtained in this paper may be significant to applications, and where one might expect assumptions to hold?
>
> **Answer**: As demonstrated by our synthetic experiments, even for $n=800, k=100$, our algorithm already offers a speedup compared to the exact alternating minimization.
>
> We would also like to emphasize that, beyond shaving off a $k$ factor and achieving nearly optimal runtime, we also provide a robust analytical framework for alternating minimization. As a powerful and widely used primitive in practice, most theoretical analyses of alternating minimization focus on exactly computing the updates. For example, if one carefully inspects the analysis in (Li et al., 2016), it heavily exploits the closed-form solution to quantify the distances between subspaces. Unfortunately, this analytical framework is not robust enough. As we have shown in our analysis, perturbing the solutions output by the solvers can lead to polynomially large errors in the final quality of the algorithm. This is in contrast with reality, where the updates can only be computed approximately. Moreover, fast solvers are used in practice to produce these approximate updates efficiently (Avron et al., 2010, Meng et al, 2014). We provide a robust analytical framework that effectively resolves this issue by using a high-accuracy solver and bounding the condition number. Thus, our work could be interpreted as a theoretical explanation of why alternating minimization works well in practice despite the presence of errors.
>
> Again, we thank you for your insightful comments, and we hope that our response may address your concerns. Please let us know if you have any further questions or concerns. We will be very happy to address them.

---

> > ### Author Response · Authors · 2024-11-22
> >
> > Haim Avron, Petar Maymounkov, and Sivan Toledo. "Blendenpik: Supercharging LAPACK's least-squares solver." SIAM Journal on Scientific Computing’10.
> >
> > Xiangrui Meng, Michael A. Saunders, and Michael W. Mahoney. "LSRN: A parallel iterative solver for strongly over-or underdetermined systems." SIAM Journal on Scientific Computing’14.
> >
> > Yuanzhi Li, Yingyu Liang, and Andrej Risteski. "Recovery guarantee of weighted low-rank approximation via alternating minimization." ICML’16.
> >
> > Yuzhou Gu, Zhao Song, Junze Yin, and Lichen Zhang. "Low rank matrix completion via robust alternating minimization in nearly linear time." ICLR’24.

---

### Author Response · Authors · 2024-11-22
**Global Response--Part 1**

We would like to express our deepest gratitude to all the reviewers, program chairs, and (senior) area chairs for their insightful comments and consideration of our paper. All reviewers agree that our theoretical contribution is sound. Additionally, since 1. weighted low rank approximation is a fundamental problem in numerical linear algebra, 2. our work improves upon the top algorithm of the well-known work (Li et al., 2016), 3. alternating minimization is a widely used and very successful algorithm in practice, 4. Our work further improves the applicability of alternating minimization by improving the running time and robustness, our work is evaluated as well-motivated and well-presented.

One concern that multiple reviewers have regarding our work is that our work lacks the experimental result or numerical illustration (Reviewer MHu4, Reviewer j2kz, and Reviewer 5iC9). Instead of using experimental results to strengthen the contribution, reviewers actually hope to see the actual reduction in running time, and our technique may indeed work well in practice. Therefore, reviewers suggest that a synthetic experiment verifying the performance of our algorithm would be sufficient.

To address reviewers' concerns, we conducted two experiments, showing the performance of our main algorithm (Algorithm 1). In both experiments, we set $M=M^*+N \in \mathbb{R}^{n \times n}$, where $M^*$ is the rank-$k$ ground truth and $N$ is a higher-rank noise matrix. We set $n=800$ and $k=100$. We generate the noise matrix $N$ as an $n\times n$ random matrix with i.i.d. Gaussian entries of zero mean and variance $1/k$. We apply the sketching matrix $S \in \mathbb{R}^{m \times n}$ with $m=150$ (we choose the CountSketch matrix (Charikar et al., 2002)) when solving for the regression problems (see Lines 7 and 10 from our Algorithm 1). We iterate the alternating minimization steps (see Line 6) for $T = 20$ times. To show the performance of our algorithm, we compare the running time and error between our Algorithm 1 and the exact solver from (Li et al., 2016).

The first experiment is the matrix completion problem. For each row of the weight matrix $W\in \mathbb{R}^{n\times n}_{\geq 0}$, we randomly select 400 entries to be equal to 1 and the remaining 400 entries to be 0. The second experiment is the general weighted low rank approximation where the weight matrix $W$ is constructed via ${\bf 1}_n{\bf 1}_n^\top+G$ for $G$ being a random matrix with standard Gaussian entries.

Below, we present our experimental results of the matrix completion problem. We generate the ground truth $M^*=XY^\top$ by a pair $X, Y\in \mathbb{R}^{n\times k}$ with random i.i.d. entries scaled by $1/\sqrt{k}$, and the distributions are Laplace, Gaussian and uniform, respectively. Our time is measured in seconds.

|Distribution|Time for exact solve|Time for approx solve|Percentage speedup|Error for exact solve|Error for approx solve|
|---------------------|----------------------------|--------------------------------|--------------------------------|-----------|-----------|
|Laplace|234|$\mathbf{205}$|$\mathbf{12.39}$%|$1.54 \times 10^{-4}$|$3.56 \times 10^{-3}$|
|Gaussian|247|$\mathbf{216}$|$\mathbf{12.55}$%|$3.85 \times 10^{-4}$|$1.33 \times 10^{-3}$|
|Uniform|248|$\mathbf{207}$|$\mathbf{16.53}$%|$1.34 \times 10^{-5}$|$4.99 \times 10^{-4}$|


We present our second experimental results as follows (recall the weight matrix is all-1’s matrix plus a noise matrix with i.i.d. standard Gaussian entries):

|Distribution|Time for exact solve|Time for approx solve|Percentage speedup|Error for exact solve|Error for approx solve|
|---------------------|----------------------------|--------------------------------|--------------------------------|-----------|-----------|
|Laplace|249|$\mathbf{222}$|$\mathbf{10.84}$%|$2.11 \times 10^{-4}$|$4.71\times 10^{-4}$|
|Gaussian|223|$\mathbf{204}$|$\mathbf{8.52}$%|$6.29 \times 10^{-5}$|$2.15 \times 10^{-4}$|
|Uniform|221|$\mathbf{215}$|$\mathbf{2.71}$%|$3.00 \times 10^{-5}$|$7.13 \times 10^{-5}$|

We note that in these two settings, our algorithm achieves a speedup compared to the algorithm of (Li et al., 2016). For the matrix completion setting, our speedup is in the range of 12%-16%, while for the dense weights regime, our speedup is roughly 10%. In 20 iterations, our approximate solver obtains similar errors as the exact solver. We expect the speedup will be more significant once the discrepancy between $n$ and $k$ is larger, as sketching is known to work well in the regime where $n\gg k$, as evidenced by the following experiment on using sketching to solve the regression $\min_{x\in \mathbb{R}^k} \\|Ax-b\\|_2^2$ for $A\in \mathbb{R}^{n\times k}$ and $b\in\mathbb{R}^n$.

---

> ### Author Response · Authors · 2024-11-22
> **Global Response--Part 2**
>
> We test the performance of regression solvers for $n=10^6, k = 500$ with a sketch size $m=5500$, and run our solver for 5 iterations. The results are as follows:
>
>
> |Distribution|Time for exact solve|Time for approx solve|Percentage speedup|Error|
> |---------------------|----------------------------|--------------------------------|-----------|-----------|
> |Gaussian|5.149|$\mathbf{3.695}$|$\mathbf{28.24}$%|0.008244|
> |Laplace|5.480|$\mathbf{3.866}$|$\mathbf{29.45}$%|0.008406|
> |Power Law|5.505|$\mathbf{4.444}$|$\mathbf{19.27}$%|0.008771|
>
> Our data matrices $A$ and response vectors $b$ are generated according to standard Gaussian, Laplace and power law distribution with $p=5$, and we measure the $\ell_\infty$ error of the solution, i.e., let $\hat x$ denote the vector outputted by the solver, we measure $\\|\hat x - x^*\\|\_\infty$ where $x^*=\arg\min_{x\in \mathbb{R}^k} \\|Ax-b\\|_2^2$. The speedup obtained here ranges from 20% to 30%, so we have strong grounds to believe that the acceleration will even be more evident when $n$ is large. However, performing weighted low rank approximation on $10^6\times 10^6$ size matrices is currently out of the scope of our computational power, we hence leave this as a future direction.
>
>
> Due to the limited time and computational resources, we only test our algorithm under these two settings. We intend to keep conducting and integrating additional experiments on weighted low rank approximation under different settings in the final version of our paper.
>
>
> Yuanzhi Li, Yingyu Liang, and Andrej Risteski. "Recovery guarantee of weighted low-rank approximation via alternating minimization." ICML’16.
>
> Moses Charikar, Kevin Chen, and Martin Farach-Colton. "Finding frequent items in data streams." ICALP’02

---

### Author Response · Authors · 2024-11-30
**Thank you for all of your valuable and insightful comments! We kindly remind you that the discussion period will soon end.**

Dear Reviewers, Program Chairs, and (Senior) Area Chairs,

Again, we would like to express our deepest gratitude for the time and effort that the reviewers, program chairs, and (senior) area chairs have dedicated to our manuscript, as well as for their insightful and valuable comments. We kindly remind you that the discussion period will soon end. We sincerely appreciate your recognition of the strengths of our work and have carefully addressed each of the weaknesses mentioned by the reviewers. Should there be any further concerns, we are eager to engage in additional discussions with the reviewers. Below, we summarize the main strengths and weaknesses of our work as assessed by the reviewers, along with our responses to address these weaknesses.

## Strengths

All four reviewers evaluated our work as having good soundness, presentation, and contribution (3: good), except that Reviewer j2kz rated our presentation as (2: fair). Specifically:

$\bullet$ Reviewer dS4q noted that our work addresses an important practical problem with applications in numerical linear algebra and machine learning, builds on the well-known alternating minimization approach, and achieves a speedup by a factor of $k$, resulting in a nearly optimal runtime.

$\bullet$ Reviewer MHu4 assessed that our work provides **significant theoretical contributions** with well-motivated and well-presented approaches, as well as a robust framework for high-precision regression solving.

$\bullet$ Reviewer j2kz and Reviewer 5iC9 commented that our work demonstrates solid theoretical results, with a framework and novel algorithm that could potentially generalize to other matrix optimization problems.

## Weaknesses and how we address them

Generally, several reviewers (Reviewer MHu4, Reviewer j2kz, and Reviewer 5iC9) raised concerns about the lack of experimental results in our paper. We addressed this primary criticism by including comprehensive numerical results demonstrating 12-16% speedups (see our global response). Specifically:

$\bullet$ Reviewer dS4q expressed concern that our assumption is stronger than those in prior works and that our key techniques were introduced in prior work (Gu et al. 2024). We clarified that our assumption is comparable to (Li et al. 2016), making the direct improvement and comparison in runtime with (Li et al. 2016) fair. Additionally, we outlined the differences between our work and (Gu et al. 2024), highlighting our novel contributions.

$\bullet$ Reviewer MHu4 questioned the connection between Problem 2.4 and the meaning of $\epsilon_0$. We provided a detailed response to these questions, and Reviewer MHu4 expressed satisfaction with our rebuttal.

$\bullet$ Reviewer j2kz raised concerns about the clarity of a specific portion of our technique overview. We clarified the algorithm presentation and addressed this unclear section.

$\bullet$ Reviewer 5iC9 mentioned that the generalization of the RIP condition required better explanation and raised questions about the dependence on problem parameters. We revised and clarified our claims regarding RIP conditions, acknowledged and corrected imprecise wording, provided detailed technical explanations about parameter dependencies and scaling behavior, and elaborated on the robustness advantages of the spectral norm over the Frobenius norm.


## Conclusion

We are confident that we have **addressed all the reviewers' comments** and eagerly anticipate their valuable responses. We remain fully committed to revising our manuscript further to ensure its suitability for ICLR 2025. Once again, we extend our gratitude to all reviewers, program chairs, and (senior) area chairs for considering our manuscript.

Best regards,

Authors.

Yuzhou Gu, Zhao Song, Junze Yin, and Lichen Zhang. "Low rank matrix completion via robust alternating minimization in nearly linear time." ICLR’24.

Yuanzhi Li, Yingyu Liang, and Andrej Risteski. "Recovery guarantee of weighted low-rank approximation via alternating minimization." ICML’16.

---

### Meta-Review · Area_Chair_2ABp · 2024-12-20

**Metareview:**

This paper introduces a new framework for weighted low-rank approximation based on approximate alternating updates. With the proposed approach, the authors show that the algorithm’s runtime is reduced from $\|W\|_0 k^2$ to $\|W\|_0 k$ , where
$k$ is the rank of the sought matrix. The paper provides a solid theoretical analysis, and the reviewers acknowledge its technical soundness, the significance of the work, and the potential for generalizing the results to other low-rank matrix estimation problems. During the rebuttal phase, the authors addressed the reviewers’ concerns and provided numerical results that corroborate their theoretical findings.  I believe that the paper offers valuable insights into an important problem and thus recommend its acceptance to ICLR.

**Additional Comments On Reviewer Discussion:**

During the rebuttal period, the authors provided numerical experiments—initially missing from the paper—that verify their theoretical results. The absence of such results was the main concern raised by Reviewer MHu4, Reviewer j2kz, and Reviewer 5iC9. The authors also offered clarifications regarding Assumption 2.4.2, addressing Reviewer dS4q’s concern about its relative strength compared to existing literature. In addition, they further explained the RIP condition and improved the presentation of certain technical aspects of the paper. Overall, I find the rebuttal convincing, and I believe that all points raised by the reviewers were sufficiently addressed."

---

### Decision · Program_Chairs · 2025-01-22

Accept (Poster)